# Comprehensive Studies on the Regulation of Type 2 Diabetes by Cucurbitane-Type Triterpenoids in *Momordica charantia* L.: Insights from Network Pharmacology and Molecular Docking and Dynamics

**DOI:** 10.3390/ph18040474

**Published:** 2025-03-27

**Authors:** Yang Niu, Peihang Li, Zongran Pang

**Affiliations:** Key Laboratory of Ethnic Medicine in Ministry of Education, School of Pharmacy, Minzu University of China, Beijing 100081, China; n15233121472@163.com (Y.N.); lipeihang2024@163.com (P.L.)

**Keywords:** *Momordica charantia* L., cucurbitane-type triterpenoids, type 2 diabetes, network pharmacology, molecular docking, molecular dynamics simulation

## Abstract

**Background/Objectives:** *Momordica charantia* L. (*M. charantia*), a widely cultivated and frequently consumed medicinal plant, is utilized in traditional medicine. Cucurbitane-type triterpenoids, significant saponin components of *M. charantia*, exhibit hypoglycemic effects; however, the underlying mechanisms remain unclear. **Methods:** This study utilized comprehensive network pharmacology to identify potential components of *M. charantia* cucurbitane-type triterpenoids that may influence type 2 diabetes mellitus (T2DM). Additionally, molecular docking and molecular dynamics studies were performed to assess the stability of the interactions between the selected components and key targets. **Results:** In total, 22 candidate active components of *M. charantia* cucurbitane-type triterpenoids and 1165 disease targets for T2DM were identified through database screening. Molecular docking and molecular dynamics simulations were conducted for five key components (Kuguacin J, 25-O-methylkaravilagenin D, Momordicine I, momordic acid, and Kuguacin S) and three key targets (AKT1, IL6, and SRC), and the results demonstrated stable binding. The experimental results indicate that the interactions between momordic acid-AKT1 and momordic acid-IL6 are stable. **Conclusions:** Momordic acid may play a crucial role in *M. charantia*’s regulation of T2DM, and AKT1 and IL6 seem to be key targets for the therapeutic action of *M. charantia* in managing T2DM.

## 1. Introduction

Type 2 diabetes mellitus (T2DM) has become a global health epidemic, affecting hundreds of millions of people worldwide and posing a significant challenge to public health systems [1]. The rising incidence of T2DM is largely attributed to modern lifestyles, including sedentary behaviors and unhealthy dietary habits, which have led to an increase in obesity and insulin resistance [2]. This condition not only impacts the quality of life of affected individuals but also places a substantial economic burden on healthcare systems due to the costs associated with long-term management and complications.

*Momordica charantia* L. *(M. charantia)*, a cucurbitaceous plant widely consumed as both food and medicine, has garnered extensive scientific attention for its anti-diabetic properties. Phytochemical analyses reveal its rich composition of cucurbitane-type triterpenoids (e.g., momordicine IV and charantoside B) [3], saponins (charantin) [4], peptides [5], and polysaccharides [6], which collectively contribute to its hypoglycemic effects. In vitro studies demonstrate that these compounds enhance glucose uptake in skeletal muscle cells via AMP-activated protein kinase (AMPK) activation [7], inhibit α-glucosidase activity [8], and protect pancreatic β-cells from oxidative stress [9]. Animal models of streptozotocin-induced diabetes further validate these mechanisms, showing significant reductions in fasting blood glucose (FBG) and improved insulin sensitivity upon *M. charantia* extract administration [10,11]. The hypoglycemic potential of *M. charantia* has been substantiated in clinical trials involving human participants, reinforcing its traditional use in diabetes management. A systematic review and meta-analysis of ten randomized controlled trials (*n* = 1045) demonstrated that *M. charantia* monotherapy significantly reduced fasting plasma glucose (−0.72 mmol/L), postprandial glucose (−1.43 mmol/L), and HbA1c (−0.26%) compared to a placebo, with sustained effects over 4–16 weeks [12]. Further supporting this, a pilot clinical study reported that the adjunctive use of 200 mL/day fresh *M. charantia* juice alongside conventional anti-diabetic drugs reduced fasting and postprandial blood glucose by 30% and 32%, respectively, in type 2 diabetic patients over 90 days, outperforming pharmacotherapy alone [13]. These findings align with mechanistic studies suggesting insulinomimetic activity through enhanced β-cell regeneration and glucose uptake modulation. While variations in preparation methods (e.g., fresh vs. dried extracts) may influence efficacy, the cumulative clinical evidence positions *M. charantia* as a viable adjunct for glycemic control.

The regulation of T2DM involves a complex interplay among insulin signaling, inflammatory pathways (e.g., NF-κB), and metabolic regulators like PPARγ and PTP1B. While existing studies emphasize *M. charantia*’s insulin secretagogue effects [14], emerging evidence suggests broader modulatory roles in lipid metabolism and adipokine regulation [15]. Recent proteomic analyses have identified novel targets, including adiponectin upregulation and TNF-α suppression [16], positioning *M. charantia* as a multitarget therapeutic agent. However, the structural basis of its bioactive components interacting with key targets remains underexplored, particularly regarding the dynamic stability of ligand–receptor complexes.

This study employed network pharmacology to evaluate the multitarget effects of active components in *M. charantia*, thereby uncovering the underlying mechanisms of their action in T2DM. Molecular docking techniques were utilized to predict the binding modes and affinities between cucurbitane-type triterpenoids from *M. charantia* and key T2DM targets. Additionally, molecular dynamics simulations were conducted to validate the stability and dynamic behavior of these binding modes, providing robust support for a deeper understanding of their molecular-level mechanisms.

## 2. Results

### 2.1. Components and Targets

By using various databases, 22 candidate active components of cucurbitane-type triterpenoids in *M. charantia* were screened and identified. The component information is presented in Table 1, and the structural formulas are illustrated in Figure 1. The SwissTargetPrediction database and the SEA search server predicted 547 and 17 targets, respectively. After eliminating duplicate targets, a total of 553 unique targets for cucurbitane-type triterpenoids were identified (Appendix A). Among these, momordic acid had the highest number of potential targets (118), followed by Kuguacin E (117), Kuguacin N, and 3, 7, 25-Trihydroxycucurbita-5, 23-dien-19-al (111). Moreover, 25-O-methylkaravilagenin D, Kuguacin C, and Kuguacin L had 110 potential targets, while Kuguacin A and Kuguacin S had 109 potential targets. In addition, T2DM-related targets were gathered from multiple databases, including 55 targets from the PharmGkb database, 81 targets from the Drugbank database, 521 targets from the OMIM database, and 587 targets from the Genecard database. After removing duplicate items, 1165 T2DM-related targets were obtained (Appendix A). The intersection of the predicted targets for cucurbitane-type triterpenoids and T2DM targets was analyzed using Venn diagrams, which identified 121 targets as potential candidates for intervention in T2DM by cucurbitane-type triterpenoids in *M. charantia* (Figure 2A).

### 2.2. Target Screening and PPI Network Analysis

To better understand the relationships and functions of the 121 intersection targets, a PPI network was constructed using the STRING database and visualized with Cytoscape 3.10.3 software (Figure 2B). This PPI network consists of 120 nodes and 1541 edges, with a PPI enrichment *p*-value of <1.0 × 10^−16^. The size and color of the nodes vary according to their degree values. The degree value represents the number of edges connected to a node and reflects its importance within the network. Among these targets, AKT1, IL6, and SRC exhibit the highest degree values of 75, 72, and 69, respectively, indicating that they are potential key targets.

To identify the key components of cucurbitane-type triterpenoids in *M. charantia* that intervene in T2DM, an interaction network was constructed involving 22 components of cucurbitane-type triterpenoids in *M. charantia* and their intersecting targets (Figure 2C). Among these 22 components, Kuguacin J (degree value = 35), 25-O-methylkaravilagenin D (degree value = 33), Momordicine I, momordic acid, and Kuguacin S (degree value = 31) were recognized as significant, along with Kuguacin N, Karavilagenin D, and Kuguacin L (degree value = 29). In conclusion, Kuguacin J, 25-O-methylkaravilagenin D, Momordicine I, momordic acid, and Kuguacin S may be significant cucurbitane-type triterpene components that play a role in interventions in T2DM.

### 2.3. Functional Enrichment Analysis of Key Targets

To gain a comprehensive understanding of the mechanism of action of cucurbitane-type triterpenoids in *M. charantia* in T2DM at the systemic level, 121 intersecting genes were analyzed using the DAVID database. During the GO enrichment analysis, 644 items were identified in Biological Processes (BPs), 72 items in Cellular Components (CCs), and 168 terms in Molecular Functions (MFs). The top 10 words in each category are illustrated using bubble charts (Figure 3A,B). The primary indicators identified included response to a xenobiotic stimulus, a plasma membrane, and identical protein binding. Furthermore, 160 KEGG terms were found to be associated with the 121 intersecting genes. The results indicated that *M. charantia* could enhance T2DM management through multiple signaling pathways. A total of 30 significant pathways are displayed in a bar chart (Figure 3C). In the KEGG analysis, most pathways were enriched by multiple target genes. Among the most prominent pathways were Lipid and Atherosclerosis (hsa05417), the AGE-RAGE signaling pathway in diabetic complications (hsa04933), and the PI3K-Akt signaling pathway (hsa04151). The diabetes-related pathways and their associated targets were selected for further verification and analysis in subsequent experiments.

### 2.4. Molecular Docking Verification

Molecular docking technology was employed to assess the binding affinities of five cucurbitane-type triterpenoids components (Kuguacin J, 25-O-methylkaravilagenin D, Momordicine I, momordic acid, and Kuguacin S) with three key target genes (AKT1, IL6, and SRC). The results indicated that the docking scores of these five cucurbitane-type triterpenoids with the three targets ranged from −6.27 kcal/mol to −7.8 kcal/mol (Table 2). For AKT1, momordic acid exhibited the strongest binding affinity, with a score of −7.3 kcal/mol. For IL6 and SRC, momordic acid also had the strongest binding affinity, with docking scores of −7.7 kcal/mol and −7.8 kcal/mol, respectively, demonstrating a robust binding affinity for these key targets. Figure 4 illustrates the binding mechanisms between the three key targets and their selected cucurbitane-type triterpene ligands. The binding pockets of AKT1, IL6, and SRC were effectively occupied by Kuguacin J, 25-O-methylkaravilagenin D, Momordicine I, momordic acid, and Kuguacin S. As depicted in Figure 4A–C, the molecular docking of momordic acid with the AKT1 protein was conducted, revealing a binding energy of −7.3 kcal/mol. Since this binding energy was less than −5 kcal/mol, it indicated that momordic acid could spontaneously and effectively bind to the AKT1 protein. Visual analysis was performed using Discovery Studio 2019 and PyMOL, revealing that momordic acid could stably bind to the cavity of the AKT1 protein and interact with surrounding amino acids. The primary interactions involved hydrogen bonds and hydrophobic interactions. Specifically, the AKT1 protein formed a stable hydrogen bond with ARG48 in momordic acid, while the hydrophobic groups in momordic acid interacted with ILE6, VAL4, and VAL45 in the AKT1 protein, which served as the main driving force behind momordic acid binding to the active site. 2.1 The molecular docking of momordic acid with the IL6 protein was conducted. The results indicated that the binding energy for this docking was −7.7 kcal/mol. This value was less than −5 kcal/mol, which suggests that momordic acid can spontaneously and effectively bind to the IL6 protein. Visual analysis was performed using Discovery Studio 2019 and PyMOL 3.1.3, revealing that momordic acid can stably bind to the cavity of the IL6 protein and interact with the surrounding amino acids. The primary interactions between momordic acid and the IL6 protein occur through hydrogen bonds and hydrophobic interactions. Specifically, the IL6 protein forms a stable hydrogen bond with ASP34 in momordic acid, while the hydrophobic groups in momordic acid interact with ARG182, LEU187, ARG30, LEU33, and LYS171 in the IL6 protein. These interactions serve as the main driving force behind momordic acid binding to the active site. The molecular docking of momordic acid with the SRC protein was carried out. The results showed that the binding energy of this docking was −7.8 kcal/mol and the binding energy was less than −5 kcal/mol, which indicated that momordic acid could spontaneously bind effectively to the SRC protein. Visual analysis was performed using Discovery Studio 2019 and PyMOL, and it was found that momordic acid could stably bind to the cavity of the SRC protein and interact with the surrounding amino acids. Momordic acid mainly docked with the SRC protein through hydrogen bonds and hydrophobic interactions. The SRC protein could form a stable hydrogen bond with ASP34; the hydrophobic groups in momordic acid could interact with LYS171, LEU33, ARG30, LEU178, and ARG182 in the SRC protein, which was the main driving force behind momordic acid binding to the active site. In conclusion, momordic acid can dock with AKT1, IL6, and SRC.

### 2.5. Molecular Dynamics Simulation and Binding Free Energy Calculation

The root mean square deviation (RMSD) is a good indicator for measuring the conformational stability of proteins and ligands, as well as the degree of deviation of atomic positions from the starting positions [27]. The smaller the deviation, the better the conformational stability [28]. Therefore, RMSD was used to evaluate the stability of the simulation system. Some studies have found that structures with a lower RMSD are more likely to become suitable candidate structures, and a model with an RMSD of 4.5 Å obviously corresponds with good prediction [29]. As shown in Figure 5A, except for the SRC–momordic acid complex system, the AKT1–momordic acid and IL6–momordic acid complex systems reached equilibrium after 10 ns and finally fluctuated around 3.2 Å and 2.5 Å. Therefore, the momordic acid small molecule exhibited high stability when binding to the AKT1 and IL6 target proteins. In addition, relevant studies have shown that the Radius of Gyration (Rg) characterizes the compactness of proteins. The lower this value, the more compact the protein structure, and the larger the value, the looser the protein structure [30]; the larger the solvent-accessible surface area (SASA) value, the larger the exposed area of the protein and the fewer the buried hydrophobic surfaces; the smaller the SASA value, the more compact the protein structure and the more buried hydrophobic surfaces there are [31]. In our experimental study, the Rg and SASA of the AKT1–momordic acid and IL6–momordic acid complex systems fluctuated relatively stably during the movement process, indicating that the AKT1–momordic acid and IL6–momordic acid complex systems remained stable and compact throughout the simulation process. Hydrogen bonds play an important role in the binding of ligands to proteins. During the dynamic process, the number of hydrogen bonds in the AKT1–momordic acid and IL6–momordic acid complex systems fluctuated between 0 and 5. In most cases, there were approximately two hydrogen bonds in the complex systems, indicating that these two complex systems had good hydrogen bond interactions (Figure 5D). RMSF can represent the flexibility of amino acid residues in proteins, which characterizes the fluctuation of each amino acid. This value indicates the stability of amino acids. The larger the RMSF value, the less stable the amino acid, and conversely, the smaller the RMSF value, the more stable the amino acid [32]. As shown in Figure 5E–H, the RMSF values of the AKT1–momordic acid and IL6–momordic acid complex systems were relatively low (mostly below 4 Å), indicating low flexibility and high stability. In conclusion, the AKT1–momordic acid and IL6–momordic acid complex systems had stable binding, and the complexes had good hydrogen bond interactions. Therefore, the momordic acid small molecule had a good binding effect with the AKT1 and IL6 target proteins.

## 3. Discussion

Our investigation elucidated the multitarget therapeutic potential of Momordica charantia-derived cucurbitane-type triterpenoids in T2DM interventions, advancing previous research by integrating network pharmacology with dynamic conformational analyses. While earlier studies primarily focused on single-target interactions or phenomenological observations of *M. charantia* extracts [33,34], our systems-level approach revealed the coordinated modulation of PI3K−AKT and AGE-RAGE signaling—a finding corroborated by recent work demonstrating pathway crosstalk in insulin sensitization [35]. Notably, the superior binding affinity of momordic acid to AKT1 (−7.3 kcal/mol) and IL6 (−7.7 kcal/mol) aligns with structural insights from SAR studies on cucurbitane derivatives [3], where hydroxyl and carboxyl moieties enhance kinase interactions through hydrogen bonding with catalytic residues like ARG48 in AKT1 [36]. This mechanistic clarity addresses a critical gap in prior computational models limited to static docking, as our 50 ns molecular dynamics simulations confirm stable ligand–receptor complexes through both polar interactions and the hydrophobic stabilization of sterically bulky regions−a dual binding mode absent in less potent analogs like kuguacin J.

The identification of IL6 as a key anti-inflammatory target introduces a paradigm shift from conventional views of its pro-diabetic role. While chronic IL6 elevation exacerbates insulin resistance [37], our data suggest context-dependent modulation where momordic acid’s stabilization of IL6 and AKT1 interactions may reconcile contradictory reports on cytokine function in metabolic regulation. This aligns with emerging evidence that low-grade inflammation modulation—rather than complete suppression—optimizes glycemic control [38]. Such nuanced immunometabolic regulation positions cucurbitane triterpenoids as superior to single-pathway inhibitors like metformin [39], which primarily targets hepatic gluconeogenesis without addressing inflammatory components [40].

From a translational standpoint, cucurbitane-type triterpenoids’ physicochemical properties and bioavailability will dictate their therapeutic formulation. Given their natural origin and historical use in traditional diets, these compounds could be optimized as standardized dietary supplements to enhance glycemic control in prediabetic or early-stage T2DM populations. For instance, bioactive fractions of *M. charantia* have already been commercialized as nutraceuticals in Asian markets [41,42], though efficacy standardization remains a challenge. However, bioavailability challenges inherent to triterpenoids—as seen with berberine and curcumin —necessitate formulation innovations [43]. Structural optimization through glycosylation or nanoparticle encapsulation (to improve intestinal absorption) could transform these compounds into pharmaceutical-grade agents, though rigorous safety assessments would be essential. Preclinical models demonstrate feasibility: saponin-rich *M. charantia* fractions reduced fasting glucose in diabetic mice via AMPK activation [44], while cucurbitane glycosides increased GLUT4 translocation in adipocytes [45]. These findings validate our computational predictions and suggest prioritizing the evaluation of momordic acid in rodent models using dose escalation protocols mirroring successful anti-diabetic natural product development [46,47].

Methodologically, our integration of KEGG pathway analysis into molecular dynamics represents a significant advancement compared to previous network pharmacology studies on *M. charantia* [48]. While Taheri et al. identified PI3K/AKT dysregulation in insulin resistance, our work uniquely maps ligand-induced conformational stabilization to pathway reactivation—a critical step for rational drug design. Notably, our findings complement recent discoveries of isomerization phenomena in cucurbitane triterpenoids during extraction, emphasizing the necessity of dynamic conformational analysis. Future studies should employ surface plasmon resonance to experimentally validate binding affinities, particularly for momordic acid’s interaction with AKT1’s PH domain, where its flexible side chains occupy a hydrophobic pocket adjacent to the catalytic cleft. Such validation would bridge the gap between computational prediction and therapeutic application, addressing a persistent limitation in phytochemical research [49].

In conclusion, this study establishes a structural and systems-level framework for repurposing cucurbitane triterpenoids as multitarget T2DM therapeutics. By elucidating both molecular interactions and pathway synergies, we provide a roadmap for developing standardized nutraceuticals or optimized pharmaceuticals. Clinical trials should explore adjunctive use with existing therapies (e.g., metformin or GLP-1 agonists), leveraging these compounds’ unique capacity to concurrently target insulin signaling and inflammation—a combinatorial strategy poised to address diabetes-associated comorbidities more effectively than current monotherapies.

Despite these advances, our study has limitations. While computational models predict therapeutic potential, in vivo validation is imperative to confirm bioavailability, tissue-specific effects, and long-term safety. Particular attention should be paid to the dose-dependent effects observed in STZ-induced diabetic mice models, as well as potential microbiota-mediated mechanisms reported in HFD-fed mice. Future work should employ diabetic rodent models to assess dose-dependent glucose-lowering effects and compare cucurbitane-type triterpenoids with existing anti-diabetic agents. Additionally, clinical trials could evaluate the synergistic effects when these compounds are co-administered with metformin or GLP-1 agonists, leveraging their multitarget profiles to enhance therapeutic outcomes. These interactions can also be confirmed experimentally using techniques such as surface plasmon resonance (SPR) or enzyme inhibition analysis. These methods can provide direct measurements of combined affinity and kinetics, which will further validate computational predictions and strengthen the credibility of the findings.

## 4. Materials and Methods

### 4.1. Collection of Component Targets

By referencing the literature and utilizing the Swiss ADME platform (http://www.swissadme.ch/, accessed on 14 January 2025) [50] to analyze drug likeness properties, we screened and identified the candidate active components of cucurbitane-type triterpenoids in *M. charantia*. The criteria for screening were set as follows: the gastrointestinal absorption was defined as “high” [51], and for a compound to be selected, it should meet the criteria of “yes” for two or more out of the five drug properties (Lipinski’s rules, Ghose’s rules, Veber’s rules, Egan’s rules, and Muegge’s rules) [52]. The selected components were then used to obtain their SMILES representations through the PubChem database (https://pubchem.ncbi.nlm.nih.gov/, accessed on 14 January 2025). Subsequently, these SMILES representations were employed in the SwissTargetPrediction database (http://www.swisstargetprediction.ch/, accessed on 14 January 2025) [53] to identify the targets of the components, while the SEA search server (https://sea.bkslab.org/, accessed on 14 January 2025) was utilized to supplement the target information. The screening criteria were set as “*Homo sapiens*” and “HUMAN”, with a *p*-Value < 0.05 [54]. Finally, the predicted targets from both databases were merged, and duplicate entries were removed to compile a list of potential targets for the candidate active components of cucurbitane-type triterpenoids in *M. charantia*.

### 4.2. Collection of Disease Targets

Using ‘type 2 diabetes’ as the keyword, T2DM-related targets were retrieved from four disease databases: PharmGkb (https://www.pharmgkb.org/, accessed on 16 January 2025) [55], Drugbank (https://go.drugbank.com/, accessed on 16 January 2025) [56], OMIM (https://www.omim.org/, accessed on 16 January 2025) [57], and GeneCards (https://www.genecards.org/, accessed on 16 January 2025) [58]. For Genecard, the screening criteria were “Protein Coding” and the median value [59]. Subsequently, the T2DM-related targets from the four databases were consolidated, and duplicate entries were eliminated.

### 4.3. Construction of PPI and CTD Networks

To investigate the potential targets of cucurbitane-type triterpenoids in the treatment of T2DM, an intersection analysis was conducted between the targets of cucurbitane-type triterpenoids and those associated with T2DM. This analysis was performed using an online platform (https://bioinfogp.cnb.csic.es/tools/venny/, accessed on 17 January 2025) to generate a Venn diagram. The intersecting targets were then imported into the STRING database (http://string-db.org/, accessed on 18 January 2025) to construct a protein–protein interaction (PPI) network for the target proteins [13]. The species *Homo sapiens* was selected, and all other parameters were set to their default values. The network was visualized using Cytoscape 3.10.3 software, and the degree value of each target was calculated using the network analysis plugin CentiScaPe 2.2. The experiment adopted a commonly used method for obtaining key targets [60,61]. By comprehensively considering betweenness, closeness, and degree, core targets were screened. Finally, based on the degree, key targets were determined, and the top three proteins with the highest degree values were selected for molecular docking studies.

### 4.4. GO and KEGG Enrichment Analysis

In order to evaluate the mechanism of action of *M. charantia* in T2DM, Gene Ontology (GO) and Kyoto Encyclopedia of Genes and Genomes (KEGG) data for key targets were retrieved using the DAVID database (https://david.ncifcrf.gov/). *p* < 0.05 was considered statistically significant. The graphs were generated using RStudio (2022.07.1, Build 554).

### 4.5. Molecular Docking

The 2D structures of small-molecule ligands were obtained from the PubChem database (https://pubchem.ncbi.nlm.nih.gov/, accessed on January 19, 2025). These 2D structures were then inputted into Chem3D 22.0.0 to generate their 3D structures, which were saved as .mol2 files. Subsequently, the RCSB PDB database (http://www.rcsb.org/, accessed on 22 January 2025) was utilized to screen for high-resolution crystal structures of protein targets to serve as molecular docking receptors. PyMOL 3.1.3 software was employed to perform dehydration and dephosphorylation operations on the protein, and the modified structure was saved as a PDB file. Molecular docking was conducted using AutoDock Vina 1.5.6 software to explore the interaction between the protein and the ligand. Autodock was used to process the structures of both the protein and the small molecule, including the hydrogenation and dehydration of the protein, as well as the hydrogenation and torsion force determination of the small-molecule ligand. The coordinates of the docking box were subsequently defined. By comparing the docking scores, the optimal conformation for molecular simulation was ultimately identified. Discovery Studio 2019 software was utilized for visual analysis of the 2D and 3D interaction diagrams between the test compound and key residues. 

### 4.6. Molecular Dynamics Simulations

To deeply explore the kinetic behavior of the protein–ligand complex, this study conducted a 100 ns molecular dynamics (MD) simulation using the Gromacs 2022 software.

In the simulation preparation stage, for the protein part, we selected the CHARMM 36 force field parameters to accurately describe the interatomic interactions within it. The topological structure of the ligand was constructed based on the GAFF2 force field parameters to ensure an accurate depiction of the molecular characteristics of the ligand.

To establish a suitable simulation environment, we employed periodic boundary conditions and placed the protein–ligand complex in a cubic box. Subsequently, the TIP3P water model was used to fill the box with water molecules, thereby simulating the solvent effect in the physiological environment.

Regarding the treatment of interactions, the electrostatic interactions were handled using the Particle Mesh Ewald (PME) algorithm, and the Verlet algorithm was applied to ensure the stability and accuracy of the simulation.

Before the formal simulation, the system was carefully equilibrated. First, 100,000 steps of the NVT (isothermal–isochoric ensemble) equilibration were carried out to stabilize the system temperature. Then, the NPT (isothermal–isobaric ensemble) equilibration was performed with a coupling constant set to 0.1 ps and an equilibration duration of 100 ps, enabling the system to reach an equilibrium state in both pressure and temperature. During this process, the van der Waals and Coulomb interactions were both calculated using a cutoff value of 1.0 nm.

Finally, under the conditions of a constant temperature of 310 K and a constant pressure of 1 bar, Gromacs 2022 software was used to conduct a 100 ns molecular dynamics simulation of the system to obtain the structural and kinetic information of the complex during the dynamic process.

## 5. Conclusions

In this study, network pharmacology and computer simulation techniques were employed to investigate the potential targets and molecular mechanisms of cucurbitane-type triterpenoids in *M. charantia* acting on T2DM. Five key cucurbitane-type triterpene components were identified: Kuguacin J, 25-O-methylkaravilagenin D, Momordicine I, momordic acid, and Kuguacin S. Additionally, three potential therapeutic targets were identified: AKT1, IL6, and SRC. The results of molecular docking and molecular dynamics simulations indicated that momordic acid may effectively bind to these key targets through hydrogen bonds and hydrophobic interactions, thereby playing a significant role in the regulation of T2DM. These findings contribute to our understanding of the potential therapeutic value of cucurbitane-type triterpenoids in the treatment or prevention of T2DM and provide a solid foundation for further research. However, a limitation of this study is that additional pharmacological and clinical investigations are necessary to validate our conclusions.

## Figures and Tables

**Figure 1 pharmaceuticals-18-00474-f001:**
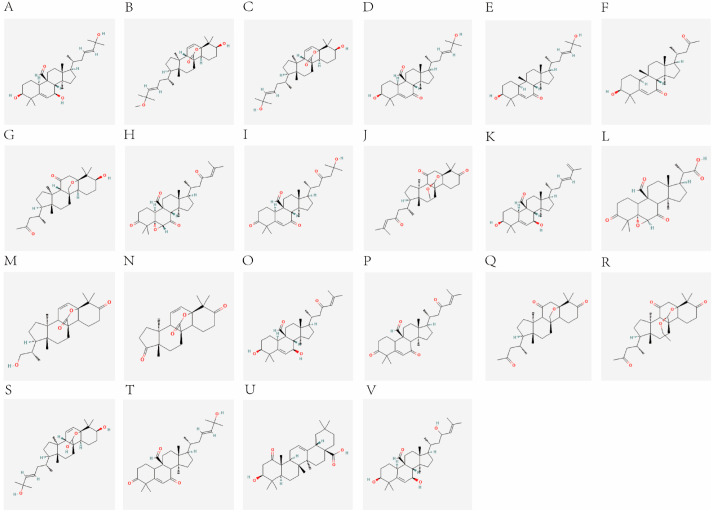
Structural formulas of 22 candidate active components of cucurbitane-type triterpenoids found in *M. charantia*. (**A**) Momordicine I. (**B**) Kuguacin J. (**C**) Kuguacin R. (**D**) Kuguacin N. (**E**) 25-O-methylkaravilagenin D. (**F**) 3, 7, 25-Trihydroxycucurbita-5, 23-dien-19-al. (**G**) Karavilagenin D. (**H**) momordic acid. (**I**) Kuguacin A. (**J**) Kuguacin B. (**K**) Kuguacin C. (**L**) Kuguacin E. (**M**) Kuguacin F. (**N**) Kuguacin H. (**O**) Kuguacin I. (**P**) Kuguacin K. (**Q**) Kuguacin L. (**R**) Kuguacin M. (**S**) Kuguacin O. (**T**) Kuguacin P. (**U**) Kuguacin Q. (**V**) Kuguacin S.

**Figure 2 pharmaceuticals-18-00474-f002:**
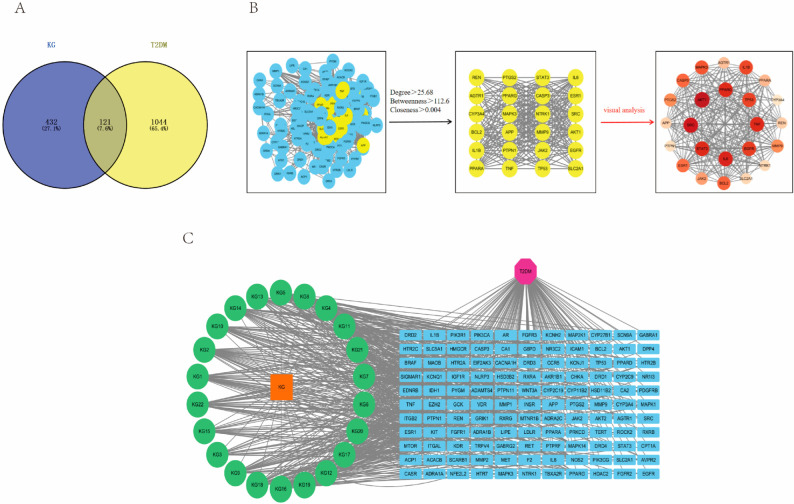
Screening of common and key targets. (**A**) Venn diagram illustrating component and disease targets. (**B**) Process of PPI network analysis. (**C**) Diagram of the component–disease–target interaction process.

**Figure 3 pharmaceuticals-18-00474-f003:**
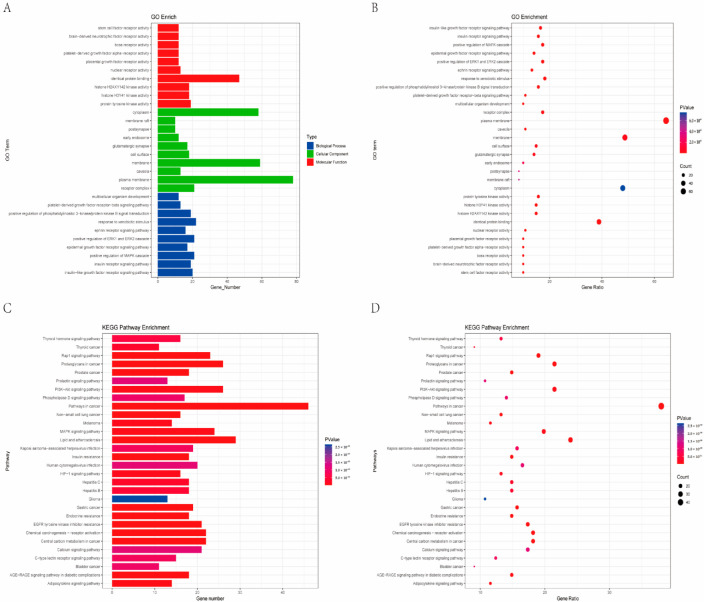
Charts for GO and KEGG enrichment analysis. (**A**) Target-enriched GO pathway histogram. (**B**) Target-enriched GO pathway bubble plot. (**C**) Target protein-enriched KEGG pathwayhistogram. (**D**) Target protein-enriched KEGG pathway bubble plot.

**Figure 4 pharmaceuticals-18-00474-f004:**
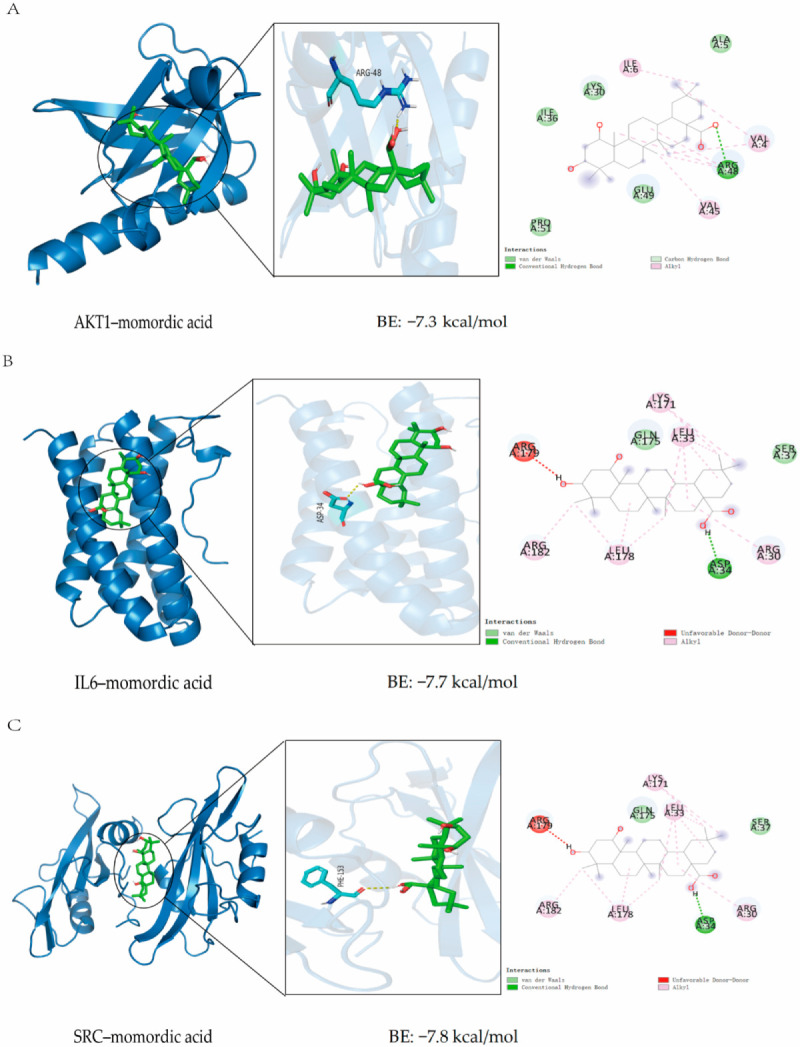
Display of molecular docking results: (**A**) AKT1–momordic acid. (**B**) IL6–momordic acid. (**C**) SRC–momordic acid.

**Figure 5 pharmaceuticals-18-00474-f005:**
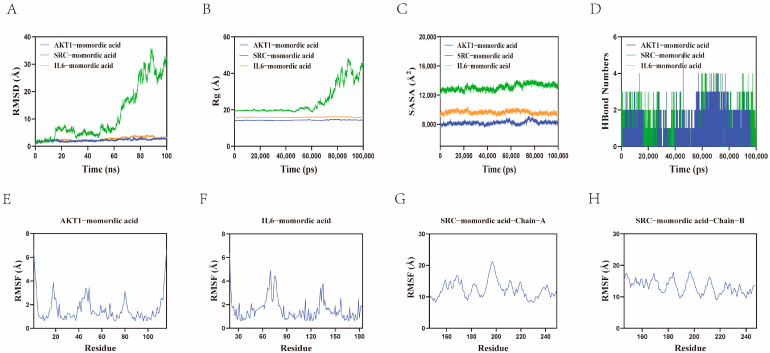
Display of molecular dynamics simulation results. (**A**) Root mean square deviation. (**B**) Radius of Gyration. (**C**) Solvent-accessible surface area. (**D**) Hydrogen bonding situation. (**E**) Root mean square fluctuation of AKT1–momordic acid. (**F**) Root mean square fluctuation of IL6–momordic acid. (**G**) Root mean square fluctuation of SRC−momordic acid−Chain A. (**H**) Root mean square fluctuation of SRC−momordic acid−Chain B.

**Table 1 pharmaceuticals-18-00474-t001:** The 22 candidate active components of cucurbitane-type triterpenoids in Momordica charantia.

NO.	Compound Name	SMILES	References
A	Momordicine I	C[C@H](CC(C=C(C)C)O)[C@H]1CC[C@@]2([C@@]1(CC[C@@]3([C@H]2[C@H](C=C4[C@H]3CC[C@@H](C4(C)C)O)O)C=O)C)C	[7,17]
B	Kuguacin J	C[C@H](C/C=C/C(=C)C)[C@H]1CC[C@@]2([C@@]1(CC[C@@]3([C@H]2[C@H](C=C4[C@H]3CC[C@@H](C4(C)C)O)O)C=O)C)C	[18]
C	Kuguacin R	C[C@H](C/C=C/C(C)(C)O)[C@H]1CC[C@@]2([C@@]1(CC[C@]34[C@H]2C=C[C@]5([C@H]3CC[C@@H](C5(C)C)O)OC4O)C)C	[19]
D	Kuguacin N	C[C@H](CC(=O)C=C(C)C)[C@H]1CC[C@@]2([C@@]1(CC[C@@]3([C@H]2[C@H](C=C4[C@H]3CC[C@@H](C4(C)C)O)O)C=O)C)C	[19]
E	25-O-methylkaravilagenin D	C[C@H](C/C=C/C(C)(C)OC)[C@H]1CC[C@@]2([C@@]1(CC[C@]34[C@H]2C=C[C@]5([C@H]3CC[C@@H](C5(C)C)O)OC4=O)C)C	[20]
F	3, 7, 25-Trihydroxycucurbita-5, 23-dien-19-al	C[C@H](C/C=C/C(C)(C)O)[C@H]1CC[C@@]2([C@@]1(CC[C@@]3([C@H]2[C@H](C=C4[C@H]3CC[C@@H](C4(C)C)O)O)C=O)C)C	[21]
G	Karavilagenin D	C[C@H](C/C=C/C(C)(C)O)[C@H]1CC[C@@]2([C@@]1(CC[C@]34[C@H]2C=C[C@]5([C@H]3CC[C@@H](C5(C)C)O)OC4=O)C)C	[22,23,24]
H	momordic acid	C[C@@]12CC[C@@H]3[C@@]([C@H]1CC=C4[C@]2(CC[C@@]5([C@H]4CC(CC5)(C)C)C(=O)O)C)(C(=O)C[C@@H](C3(C)C)O)C	[25]
I	Kuguacin A	C[C@H](C/C=C/C(C)(C)O)[C@H]1CC[C@@]2([C@@]1(CC[C@@]3([C@H]2C(=O)C=C4[C@H]3CC[C@@H](C4(C)C)O)C=O)C)C	[22,23]
J	Kuguacin B	C[C@H](C/C=C/C(C)(C)O)[C@H]1CC[C@@]2([C@@]1(CC[C@@]3([C@H]2C(=O)C=C4[C@H]3CC[C@@H](C4(C)C)O)C)C)C	[22,23]
K	Kuguacin C	C[C@H](CC(=O)C)[C@H]1CC[C@@]2([C@@]1(CC[C@@]3([C@H]2C(=O)C=C4[C@H]3CC[C@@H](C4(C)C)O)C)C)C	[22,23]
L	Kuguacin E	C[C@H](CC(=O)C)[C@H]1CC[C@@]2([C@@]1(CC[C@]34[C@H]2C(=O)C[C@]5([C@H]3CC[C@@H](C5(C)C)O)OC4)C)C	[22,23]
M	Kuguacin F	C[C@H](CC(=O)C=C(C)C)[C@H]1CC[C@@]2([C@@]1(CC[C@@]3([C@H]2C(=O)[C@H]4[C@@]5([C@H]3CCC(=O)C5(C)C)O4)C=O)C)C	[19,22]
N	Kuguacin H	C[C@H](CC(=O)CC(C)(C)O)[C@H]1CC[C@@]2([C@@]1(CC[C@@]3([C@H]2C(=O)C=C4[C@H]3CCC(=O)C4(C)C)C=O)C)C	[19,22,26]
O	Kuguacin I	C[C@H](CC(=O)C=C(C)C)[C@H]1CC[C@@]2([C@@]1(CC[C@]34C2C(=O)C[C@]5(C3CCC(=O)C5(C)C)O[C@H]4OC)C)C	[19,26]
P	Kuguacin K	C[C@@H]([C@H]1CC[C@@]2([C@@]1(CC[C@@]3(C2C(=O)[C@@H]4[C@@]5(C3CCC(=O)C5(C)C)O4)C=O)C)C)C(=O)O	[19,26]
Q	Kuguacin L	C[C@H](CO)[C@H]1CC[C@@]2([C@@]1(CC[C@]34C2C=C[C@]5(C3CCC(=O)C5(C)C)OC4=O)C)C	[19,26]
R	Kuguacin M	C[C@@]12CCC(=O)[C@]1(CC[C@]34C2C=C[C@]5(C3CCC(=O)C5(C)C)OC4=O)C	[19,26]
S	Kuguacin O	C[C@H](CC(=O)C=C(C)C)[C@H]1CC[C@@]2([C@@]1(CC[C@@]3(C2C(=O)C=C4C3CCC(=O)C4(C)C)C=O)C)C	[19]
T	Kuguacin P	C[C@H](CC(=O)C)[C@H]1CC[C@@]2([C@@]1(CC[C@]34C2C(=O)C[C@]5(C3CCC(=O)C5(C)C)OC4)C)C	[19]
U	Kuguacin Q	CCO[C@H]1[C@@]23CC[C@@]4([C@H](CC[C@]4(C2C(=O)C[C@]5(C3CCC(=O)C5(C)C)O1)C)[C@H](C)CC(=O)C)C	[19]
V	Kuguacin S	C[C@H](C/C=C/C(C)(C)O)[C@H]1CC[C@@]2([C@@]1(CC[C@@]3(C2C(=O)C=C4C3CCC(=O)C4(C)C)C=O)C)C	[19]

**Table 2 pharmaceuticals-18-00474-t002:** Molecular docking results for cucurbitane-type triterpene components and core targets.

Compound Name	Affinity (kcal/mol)		
	AKT1	IL6	SRC
25-O-methylkaravilagenin D	−6.53	−7.30	−7.40
Kuguacin J	−6.27	−6.73	−7.23
Kuguacin S	−6.27	−7.33	−7.53
momordic acid	−7.30	−7.70	−7.80
Momordicine I	−6.53	−7.30	−7.40

## Data Availability

The original contributions presented in the study are included in the Appendix A.

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
