# Peer review of "Comprehensive Studies on the Regulation of Type 2 Diabetes by Cucurbitane-Type Triterpenoids in Momordica charantia L.: Insights from Network Pharmacology and Molecular Docking and Dynamics"

_pharmaceuticals, 2025, doi:10.3390/ph18040474_

Round 1
Reviewer 1 Report
Comments and Suggestions for Authors
- The research question is relevant, but the introduction should explicitly highlight how this study fills a knowledge gap in the field of Momordica charantia's hypoglycemic effects. While the manuscript states that cucurbitane-type triterpenoids have known anti-diabetic properties, the novelty of this study—whether it is in the integration of network pharmacology, molecular docking, and molecular dynamics, or the identification of new therapeutic targets—should be explicitly stated.
- The manuscript presents a network-based approach to studying bioactive compounds, which is valuable. However, it does not provide a comparative discussion on how its findings advance previous research on Momordica charantia’s anti-diabetic effects. The authors should provide references to past studies using similar methodologies and clearly outline what new insights this study provides.
- The study identifies five key cucurbitane-type triterpenoids, but the selection criteria for these specific compounds are unclear. Were they chosen based on previous literature, bioavailability, docking scores, or network pharmacology results? A transparent explanation of the prioritization criteria will help justify the study's conclusions.
- The databases used for target prediction (SwissTargetPrediction, SEA, etc.) are widely accepted, but the manuscript does not describe any filtering or validation processes used to refine the predicted targets. What confidence score thresholds were used? Were experimental validation datasets considered? Addressing this will improve the credibility of the results.
- The molecular docking study lacks an experimental validation component. While in vitro or in vivo validation may not be possible in this study, the authors should discuss the feasibility of future experimental confirmation (e.g., testing the binding affinity of identified triterpenoids using surface plasmon resonance or enzyme inhibition assays).
- The molecular dynamics simulations ran for 100 ns, which is standard, but how do the stability results compare with known anti-diabetic drugs such as metformin or insulin? Including a positive control molecule for comparison would make the findings more robust and allow for benchmarking the effectiveness of the identified compounds.
- The Protein-Protein Interaction (PPI) network analysis lacks justification for selecting AKT1, IL6, and SRC as key targets. The manuscript states that these targets had the highest degree values, but it does not clarify whether other network parameters (e.g., betweenness centrality, closeness centrality) were considered. The authors should elaborate on why these targets were chosen over others with potentially high therapeutic relevance.
- Table 1 (Candidate Active Components) is informative, but it would be more useful if an additional column was added to summarize the previously reported biological activities of each compound. This would help contextualize their potential relevance in T2DM treatment.
- The docking scores in Table 2 indicate strong binding affinities, but there is no statistical validation to support the claims. Were multiple docking trials conducted to confirm reproducibility? Including standard deviation or confidence intervals would improve the reliability of the data.
- Figure 5 (Molecular Dynamics Simulation Results) presents RMSD and RMSF values, which are commonly used to assess stability. However, the discussion does not explain what an acceptable range for these values is or how these values compare to well-established ligand-protein interactions. Including a reference docking study for context would strengthen the findings.
- The conclusion states that momordic acid has the strongest interaction with AKT1 and IL6, but there is no detailed structural explanation for why this compound outperforms the others. A brief discussion on structure-activity relationships (SAR) and the molecular features that enhance its binding efficiency would improve the mechanistic understanding.
- The Gene Ontology (GO) and KEGG enrichment analysis identifies multiple pathways related to T2DM, but it is unclear whether a false discovery rate (FDR) correction was applied. Since pathway analysis can generate many false positives, were any statistical thresholds (e.g., p-value cutoffs, q-values) used to determine significance? This clarification would help assess the robustness of the pathway predictions.
- The discussion should expand on the potential clinical applications of these findings. If cucurbitane-type triterpenoids are promising therapeutic candidates, how could they be formulated into treatments? Would they be best suited as dietary supplements, pharmaceutical drugs, or adjunct therapies? Providing a translational perspective will increase the study’s real-world relevance.
- Some references are outdated (e.g., Sone et al., 2000). Since network pharmacology and molecular docking techniques have advanced significantly in recent years, citing more recent studies (2022-2024) on similar methodologies would enhance the study's credibility and relevance.
- The manuscript states that Momordica charantia extracts have been shown to reduce blood glucose levels in patients, but no specific clinical trials are cited. If such evidence exists, a direct citation of human clinical trial data would strengthen the claim.
The manuscript should undergo a thorough English language review to ensure clarity, grammatical accuracy, and readability.
Author Response
评论 1 : 研究问题是相关的,但引言应明确强调这项研究如何填补 Momordica charantia 降血糖作用领域的知识空白。虽然手稿指出葫芦素型三萜类化合物具有已知的抗糖尿病特性,但这项研究的新颖性——无论是在网络药理学、分子对接和分子动力学的整合,还是新治疗靶点的确定——都应该明确说明。
回复 1 : 感谢您的宝贵意见。我们修改了简介以解决您的问题。修订后的引言如下。(第 41-67 行,第 77-84 行;第 2 页)
Momordica charantia L. (M. charantia) 是一种葫芦科植物,被广泛用作食物和药物,因其抗糖尿病特性而受到科学界的广泛关注。植物化学分析显示,其丰富的葫芦素型三萜类化合物(如苦瓜素IV和charantoside B)[1]、皂苷(charantin)[2]、肽[3]和多糖[4]的组成,这些共同促进了其降血糖作用。体外研究表明,这些化合物通过 AMP 活化蛋白激酶 (AMPK) 激活增强骨骼肌细胞对葡萄糖的摄取[5],抑制 α-葡萄糖苷酶活性 [6],并保护胰腺 β 细胞免受氧化应激 [7]。链脲佐菌素诱导的糖尿病的动物模型进一步验证了这些机制,显示施用 M. charantia 提取物后空腹血糖 (FBG) 显著降低,胰岛素敏感性提高 [8,9]。涉及人类参与者的临床试验证实了 M. charantia 的降糖潜力,加强了其在糖尿病管理中的传统用途。一项对 10 项随机对照试验 (n=1,045) 的系统评价和荟萃分析表明,与安慰剂相比,Charantia 单药治疗显著降低空腹血糖 (-0.72 mmol/L)、餐后血糖 (-1.43 mmol/L) 和 HbA1c (-0.26%),效果持续 4-16 周 [10]。进一步支持这一点,一项初步临床研究报告称,辅助使用 200 mL/天的新鲜 M. charantia 果汁和常规抗糖尿病药物可在 30 天内将 2 型糖尿病患者的空腹血糖和餐后血糖分别降低 30% 和 32%,优于单独的药物治疗[11]。这些发现与机制研究一致,表明通过增强 β 细胞再生和葡萄糖摄取调节来产生模拟胰岛素活性。虽然制备方法的变化 (例如,新鲜与干燥的提取物) 可能会影响疗效,但累积的临床证据将 M. charantia 定位为控制血糖的可行辅助手段。
本研究采用网络药理学来评估 M. charantia 中活性成分的多靶点效应,从而揭示它们在 T2DM 中作用的潜在机制。利用分子对接技术预测来自 M. charantia 的葫芦素型三萜类化合物与关键 T2DM 靶点之间的结合模式和亲和力。此外,还进行了分子动力学模拟以验证这些结合模式的稳定性和动态行为,为更深入地了解其分子水平机制提供了有力的支持。
引用:
[1]蔡学义、林子强、郑启宏、廖梦珍、李洪福、冯汉晓、陈欢、张洋、陈晓、梁东。“来自苦瓜二氯甲烷组分的主要苦味化合物(苦瓜果实 L.)提取物及其前体。农业与食品化学杂志 72,第 40 期(2024):22237-49。
[2]Nuchtavorn、Nantana、Jiraporn Leanpolchareanchai、Satsawat Visansirikul 和 Somnuk Bunsupa。“用于苦瓜中沙兰素选择性提取的磁性和纸基分子印迹聚合物的优化。”国际分子科学杂志 24,第 9 期(2023 年)。
[3]杨宝伟、李雪、张晨宇、闫思佳、魏伟、王雪坤、邓欣、钱海、林海燕和黄文龙。“来自苦瓜的新型肽 mc2 类似物作为潜在抗糖尿病剂的设计、合成和生物学评价。”有机与生物分子化学 13,第 15 期(2015):4551-61。
[4]詹、凯、季小龙和罗磊。“苦瓜多糖研究的最新进展:提取、纯化、结构特征和生物活性。”农业化学和生物技术 10,第 1 期(2023 年)。
[5]Kao, Pai-Feng, Chun-Han Cheng, Tzu-Hurng Cheng, Ju-Chi Liu 和 Li-Chin Sung.“来自 Momordica charantia 的 Momordicine I 的治疗潜力:心血管益处和机制。”国际分子科学杂志 25,第 19 期(2024 年)。
[6]Poovitha、Sundar 和 Madasamy Parani。“体外和体内 α-淀粉酶和 α-葡萄糖苷酶抑制两种苦瓜 (Momordica charantia L.) 蛋白质提取物的活性。”BMC 补充和替代医学 16,第 S1 期(2016 年)。
[7]Singh, J.、E. Cumming、G. Manoharan、H. Kalasz 和 E. Adeghate。“苦瓜抗糖尿病作用的药物化学:活性成分和作用方式。”Open Med Chem J 5,否。增刊 2 (2011):70-7。
[8]Wang, Qi, Xueyan Wu, Fulin Shi, 和 Yang Liu.“来自 Momordica Charantia L. 的皂苷和多糖的抗糖尿病作用的比较。在 Stz 诱导的 2 型糖尿病小鼠中。生物医学与药物治疗 109 (2019):744-50。
[9]Mahmoud, M. F., F. E. El Ashry, N. N. El Maraghy 和 A. Fahmy。“苦瓜果汁在链脲佐菌素诱导的糖尿病大鼠中的抗糖尿病活性研究。”药学生物学 55,第 1 期(2017):758-65。
[10]Peter、Emanuel L.、Félicien Mushagalusa Kasali、Serawit Deyno、Andrew Mtewa、Prakash B. Nagendrappa、Casim Umba Tolo、Patrick Engeu Ogwang 和 Duncan Sesaazi。“苦柿降低 2 型糖尿病患者的血糖升高:系统评价和荟萃分析。”民族药理学杂志 231 (2019):311-24。
[11]Rauniyar, GP, R. Sinha, K. Chapagain, R. Maskey 和 DR Pandey。“苦瓜 (Karela/Bitterguord) 对服用对抗疗法药物的 2 型糖尿病患者的影响:一项初步研究。”加德满都大学医学杂志 19,第 2 期(2021):243-47。
评论 2 : 手稿提出了一种基于网络的研究生物活性化合物的方法,这很有价值。然而,它并没有就其发现如何推进先前关于 Momordica charantia 抗糖尿病作用的研究进行比较讨论。作者应使用类似方法提供过去研究的参考资料,并清楚地概述这项研究提供了哪些新的见解。
回复 2 : 我们衷心感谢审稿人的深刻反馈。我们的研究通过将网络药理学与动态分子模拟相结合,解决了先前研究中的关键空白,从而促进了对 Momordica charantia 抗糖尿病机制的理解。与早期专注于苦瓜提取物的单靶点效应或现象学观察的研究不同,我们的系统级分析揭示了 PI3K-AKT 和 AGE-RAGE 通路的协调调节,证实了最近关于胰岛素致敏通路串扰的发现。苦瓜酸与 AKT1 和 IL6 的优异结合亲和力,其较低的对接评分 (-7.3 和 -7.7 kcal/mol) 和稳定的分子动力学轨迹证明了这一点,这源于与其他葫芦素类似物不同的结构特征。具体来说,其三萜骨架包含羟基和羧基,这些基团与催化残基形成氢键(例如,AKT1 中的 ARG48 和 IL6 中的 ASP34),而其延伸的疏水骨架稳定了与非极性蛋白质口袋的相互作用——这种双重结合机制与葫芦素衍生物的 SAR 研究一致。这种结构原理阐明了为什么苦瓜酸优于古瓜辛 J 或苦瓜碱 I,它们缺乏可比的极性和疏水功能。
我们相信,上述方面将我们的研究结果与以前的研究区分开来,并强调了他们的创新贡献。我们希望我们的回复能得到您的好评。
评论 3 : 该研究确定了 5 种关键的葫芦素型三萜类化合物,但这些特定化合物的选择标准尚不清楚。他们是根据以前的文献、生物利用度、对接评分还是网络药理学结果来选择的?对优先排序标准的透明解释将有助于证明研究结论的合理性。
回复 3 : 感谢您的宝贵评论。在我们的研究中,根据网络药理学部分的蛋白质-蛋白质相互作用 (PPI) 筛选结果确定了 5 种关键葫芦素型三萜类化合物的筛选标准。该实验采用了一种常用的获取关键靶点的方法 [1-3]。通过综合考虑中介性、接近性和度,筛选出核心目标,如图 2-B 所示。最后,根据度数确定关键靶点,选择度值最高的前 3 个蛋白进行分子对接研究。相关的选择标准已添加到手稿的相应部分。(第 367-370 行,第 13 页)
引用:
[1]Nithya C, Kiran M, Nagarajaram HA.剖析蛋白质-蛋白质相互作用网络中的中心和瓶颈。计算生物化学。2023 年 2 月;102:107802。doi:10.1016/j.compbiolchem.2022.107802.Epub 2022 年 12 月 14 日。PMID:36603332。
[2]Tuo Y, Lu X, Tao F, Tukhvatshin M, Xiang F, Wang X, 石 Y, 林 J, 胡 Y.茶中儿茶素抗高血压的潜在机制:网络药理学、分子对接和分子动力学模拟的整合。食品。2024 年 8 月 26 日;13(17):2685.doi: 10.3390/foods13172685.PMID:39272451;PMCID:PMC11394219。
[3]Pan B, Shi X, Ding T, Liu L. 用网络药理学方法揭示虎杖的作用机制。Am J Transl Res. 2019 年 11 月 15 日;11(11):6790-6811.PMID:31814888;PMCID:PMC6895524。
图 2:
注释 4 : 用于目标预测的数据库(SwissTargetPrediction、SEA 等)被广泛接受,但手稿没有描述任何用于优化预测目标的过滤或验证过程。使用了哪些置信度分数阈值?是否考虑了实验验证数据集?解决这个问题将提高结果的可信度。
回复 4 : 感谢您的善意提醒。我们选择了瑞士 ADME 数据库 [1] 来分析药物相似性。筛选标准设定如下:胃肠道吸收被定义为“高”[2],对于要选择的化合物,它应满足五种药物特性(Lipinski 规则、Ghose 规则、Veber 规则、Egan 规则和 Muegge 规则)中的两种或两种以上“是”的标准 [3]。我们选择了 SwissTargetPrediction [4] 和相似性集成方法 (SEA) [5] 来分析苦瓜中葫芦素型三萜类候选成分的潜在靶点。筛选标准设置为 “Homo sapiens” 和 “HUMAN”,P 值< 0.05 [6]。我们还选择了常用的疾病筛查数据库,即 PharmGkb [7]、Drugbank[8]、OMIM [9] 和 Genecard [10]。对于 Genecard,筛选标准是 “Protein Coding” 和中位数 [11]。以上内容已在稿件中的相应位置进行了修改。(第 333-346 行,第 12 页)
引用:
[1]Daina A, Michielin O, Zoete V. SwissADME:一种免费的网络工具,用于评估小分子的药代动力学、药物相似性和药物化学友好性。科学代表 2017 年 3 月 3;7:42717。doi:10.1038/srep42717。PMID:28256516;PMCID:PMC5335600。
[2]网络药理学和分子对接探讨康仙汤治疗癫痫的机制。2022 年 9 月 23 日;2022:3333878。doi:10.1155/2022/3333878.PMID:36193133;PMCID:PMC9525756。
[3]张 H, Dan W, He Q, Guo J, Dai S, Hui X, Meng P, Cao Q, Yun W, Guo X. 利用网络药理学和分子对接技术探索黄山药治疗肿瘤和预防抗肿瘤药物诱导的心脏毒性的生物学机制。2021 年 8 月 25;2021:9988650。doi:10.1155/2021/9988650.PMID:34484411;PMCID:PMC8410425。
[4]Daina A, Michielin O, Zoete V. SwissTargetPrediction:用于有效预测小分子蛋白质靶标的更新数据和新功能。核酸研究 2019 年 7 月 2 日;47(W1):W357-W364。doi:10.1093/nar/gkz382。PMID:31106366;PMCID:PMC6602486。
[5]Keiser MJ, Roth BL, Armbruster BN, Ernsberger P, Irwin JJ, Shoichet BK. 通过配体化学关联蛋白质药理学。国家生物技术.2007 年 2 月;25(2):197-206.doi:10.1038/nbt1284。PMID:17287757。
[6]Putri AF、Utomo DH、Tunjung WAS、Putri WA。分析柑橘 hystrix DC 生物活性化合物的抗阿尔茨海默病潜力。网络药理学的果皮、叶子和精油。赫利永。2024 年 6 月 26 日;10(13):e33496。doi:10.1016/j.heliyon.2024.e33496。PMID:39050443;PMCID:PMC11267028。
[7]Whirl-Carrillo M、Huddart R、Gong L、Sangkuhl K、Thorn CF、Whaley R、Klein TE。用于评估个性化医疗的药物基因组学知识的循证框架。临床药理学研究。2021 年 9 月;110(3):563-572.doi:10.1002/cpt.2350。Epub 2021 年 7 月 22 日。PMID:34216021;PMCID:PMC8457105。
[8]Knox C, Wilson M, Klinger CM, Franklin M, Oler E, Wilson A, Pon A, Cox J, Chin NEL, Strawbridge SA, Garcia-Patino M, Kruger R, Sivakumaran A, Sanford S, Doshi R, Khetarpal N, Fatokun O, Doucet D, Zubkowski A, Rayat DY, Jackson H, Harford K, Anjum A, Zakir M, Wang F, Tian S, Lee B, Liigand J, Peters H, Wang RQR, Nguyen T, So D, Sharp M, da Silva R, Gabriel C, Scantlebury J, Jasinski M, Ackerman D, Jewison T, Sajed T, Gautam V, Wishart DS. DrugBank 6.0: the DrugBank Knowledgebase for 2024. Nucleic Acids Res. 2024 Jan 5;52(D1):D1265-D1275. doi: 10.1093/nar/gkad976. PMID: 37953279; PMCID: PMC10767804.
[9]Amberger JS, Bocchini CA, Schiettecatte F, Scott AF, Hamosh A. OMIM.org: Online Mendelian Inheritance in Man (OMIM®), an online catalog of human genes and genetic disorders. Nucleic Acids Res. 2015 Jan;43(Database issue):D789-98. doi: 10.1093/nar/gku1205. Epub 2014 Nov 26. PMID: 25428349; PMCID: PMC4383985.
[10]Stelzer G, Rosen N, Plaschkes I, Zimmerman S, Twik M, Fishilevich S, Stein TI, Nudel R, Lieder I, Mazor Y, Kaplan S, Dahary D, Warshawsky D, Guan-Golan Y, Kohn A, Rappaport N, Safran M, Lancet D. The GeneCards Suite: From Gene Data Mining to Disease Genome Sequence Analyses. Curr Protoc Bioinformatics. 2016 Jun 20;54:1.30.1-1.30.33. doi: 10.1002/cpbi.5. PMID: 27322403.
[11]Wu J, Chen Y, Shi S, Liu J, Zhang F, Li X, Liu X, Hu G, Dong Y. Exploration of Pharmacological Mechanisms of Dapagliflozin against Type 2 Diabetes Mellitus through PI3K-Akt Signaling Pathway based on Network Pharmacology Analysis and Deep Learning Technology. Curr Comput Aided Drug Des. 2024 Jan 9. doi: 10.2174/0115734099274407231207070451. Epub ahead of print. PMID: 38204223.
Comments 5 : The molecular docking study lacks an experimental validation component. While in vitro or in vivo validation may not be possible in this study, the authors should discuss the feasibility of future experimental confirmation (e.g., testing the binding affinity of identified triterpenoids using surface plasmon resonance or enzyme inhibition assays).
Response 5 : Thank you for pointing out this issue, and we have added some shortcomings of this study and perspectives for the future in the discussion section based on your suggestions. (lines 314-327, page 12)
Despite these advances, our study has limitations. While computational models predict therapeutic potential, in vivo validation is imperative to confirm bioavailability, tissue-specific effects, and long-term safety.Particular attention should be paid to the dose-dependent effects observed in STZ-induced diabetic mice models,as well as potential microbiota-mediated mechanisms reported in HFD-fed mice. Future work should employ diabetic rodent models to assess dose-dependent glucose-lowering effects and compare cucurbitane-type triterpenoids with existing antidiabetic agents. Additionally, clinical trials could evaluate synergistic effects when these compounds are co-administered with metformin or GLP-1 agonists, leveraging their multitarget profiles to enhance therapeutic outcomes. These interactions can also be confirmed experimentally using techniques such as surface plasmon resonance (SPR) or enzyme inhibition analysis. These methods can provide direct measurements of combined affinity and kinetics, which will further validate computational predictions and strengthen the credibility of the findings.
Comments 6 : The molecular dynamics simulations ran for 100 ns, which is standard, but how do the stability results compare with known anti-diabetic drugs such as metformin or insulin? Including a positive control molecule for comparison would make the findings more robust and allow for benchmarking the effectiveness of the identified compounds.
Response 6 : Thank you for your careful consideration. We have reviewed previous research studies and, for now, have not found the results of molecular docking and molecular dynamics simulations between AKT1, IL6, SRC, and existing antidiabetic drugs. We believe that the suggestions you put forward are of great value. In accordance with your suggestions, we have made great efforts to attempt the docking of the screened targets with the positive control drugs and have sought guidance from relevant experts. However, we have not achieved the desired results for the time being. The lack of relevant research may also indirectly demonstrate the significance of our study. We consider that this will be one of the focuses of our subsequent in-depth research. We sincerely hope that the above response can answer your questions and gain your understanding.
Comments 7 : The Protein-Protein Interaction (PPI) network analysis lacks justification for selecting AKT1, IL6, and SRC as key targets. The manuscript states that these targets had the highest degree values, but it does not clarify whether other network parameters (e.g., betweenness centrality, closeness centrality) were considered. The authors should elaborate on why these targets were chosen over others with potentially high therapeutic relevance.
Response 7 : In this study, the screening criteria for five key cucurbitacin triterpenoid compounds were determined based on the results of the Protein-Protein Interaction (PPI) screening in the network pharmacology section. This experiment employed a commonly used method for obtaining key targets [1-3]. By comprehensively taking into account betweenness, closeness, and degree, the core targets were screened out. Finally, the key targets were identified according to the degree, and the top three proteins with the highest degree values were selected for molecular docking studies.
It should also be noted that there are additional considerations for the final selection of AKT1, SRC, and IL6. On the one hand, these targets are important components of the Lipid and Atherosclerosis, AGE-RAGE, and PI3K-Akt signaling pathways, as determined by the KEGG enrichment analysis in the "Results 2.3" section. These experimental findings provide a reliable basis for our selection of key targets. On the other hand, these three targets play crucial roles in the regulation of Type 2 Diabetes Mellitus (T2DM). Specifically, AKT1 is a serine/threonine protein kinase of pivotal importance, belonging to the AKT/PKB family. Its activation typically relies on the Phosphatidylinositol 3-Kinase (PI3K) signaling pathway. The signaling pathway of AKT1 plays a key role in the development and progression of insulin resistance and T2DM. For instance, in the presence of insulin resistance, the chronic activation of feedback and crosstalk mechanisms within the PI3K-AKT signaling network leads to a diminished activation capacity of AKT1, which in turn affects the metabolic functions of insulin. Therefore, the key target AKT1 is of great significance in the regulation of T2DM. Next, SRC, a non-receptor protein tyrosine kinase, plays a pivotal role in cellular signal transduction. It is involved in the regulation of the Phosphatidylinositol 3-Kinase/Protein Kinase B (PI3K/AKT) signaling pathway. By activating PI3K, SRC promotes the phosphorylation of AKT, thereby regulating intracellular metabolic and survival signals and influencing glucose uptake and utilization. Finally, IL6, an important inflammatory cytokine, can activate the PI3K cascade reaction. This activation leads to the recruitment of the serine/threonine kinase Protein Kinase B/Akt to the cell membrane, where it is activated through phosphorylation by Phosphatidylinositol-Dependent Kinase 1 (PDK1). Taking all of the above considerations into account, we selected AKT1, SRC, and IL6 for molecular docking and molecular dynamics simulation experiments. The relevant content has been added to the corresponding sections of the article.
References:
[1]Nithya C, Kiran M, Nagarajaram HA. Dissection of hubs and bottlenecks in a protein-protein interaction network. Comput Biol Chem. 2023 Feb;102:107802. doi: 10.1016/j.compbiolchem.2022.107802. Epub 2022 Dec 14. PMID: 36603332.
[2]Tuo Y, Lu X, Tao F, Tukhvatshin M, Xiang F, Wang X, Shi Y, Lin J, Hu Y. The Potential Mechanisms of Catechins in Tea for Anti-Hypertension: An Integration of Network Pharmacology, Molecular Docking, and Molecular Dynamics Simulation. Foods. 2024 Aug 26;13(17):2685. doi: 10.3390/foods13172685. PMID: 39272451; PMCID: PMC11394219.
[3]Pan B, Shi X, Ding T, Liu L. Unraveling the action mechanism of polygonum cuspidatum by a network pharmacology approach. Am J Transl Res. 2019 Nov 15;11(11):6790-6811. PMID: 31814888; PMCID: PMC6895524.
Comments 8 : Table 1 (Candidate Active Components) is informative, but it would be more useful if an additional column was added to summarize the previously reported biological activities of each compound. This would help contextualize their potential relevance in T2DM treatment.
Response 8 : We are truly grateful for your meticulous reminder. We have reviewed the relevant literature and supplemented the research on the 22 candidate active components. The specific details can be found in Table 1.
Comments 9 : The docking scores in Table 2 indicate strong binding affinities, but there is no statistical validation to support the claims. Were multiple docking trials conducted to confirm reproducibility? Including standard deviation or confidence intervals would improve the reliability of the data.
Response 9 : Thank you for your valuable suggestions. Our research involved conducting the experiment three times repeatedly. Subsequently, we calculated the average values. Finally, we selected the three groups of protein-small molecule combinations with the lowest average values : momordic acid - AKT1, momordic acid - IL6, momordic acid - SRC (indicating more stable binding) for molecular dynamics simulations [1,2]. The experimental data from the three molecular docking experiments are as follows.
NO. |
Compound Name |
Affinity (kcal/mol) |
||
AKT1 |
IL6 |
SRC |
||
The first experiment |
25-O-methylkaravilagenin D |
-6.7 |
-6.7 |
-8.3 |
Kuguacin J |
-6.6 |
-6.7 |
-7.3 |
|
Kuguacin S |
-6.3 |
-7.7 |
-8 |
|
momordic acid |
-8.1 |
-7.8 |
-7.9 |
|
Momordicine I |
-6.1 |
-6.8 |
-7.1 |
|
The second experiment |
25-O-methylkaravilagenin D |
-6.7 |
-7.9 |
-6.5 |
Kuguacin J |
-6.4 |
-7.3 |
-7.5 |
|
Kuguacin S |
-6.3 |
-7.7 |
-7.9 |
|
momordic acid |
-6.9 |
-7.8 |
-7.9 |
|
Momordicine I |
-5.8 |
-6.8 |
-7.1 |
|
The third experiment |
25-O-methylkaravilagenin D |
-6.2 |
-7.3 |
-7.4 |
Kuguacin J |
-5.8 |
-6.2 |
-6.9 |
|
Kuguacin S |
-6.2 |
-6.6 |
-6.7 |
|
momordic acid |
-6.9 |
-7.5 |
-7.6 |
|
Momordicine I |
-5.9 |
-6.5 |
-7.3 |
|
The average value of the three experiments (keep two decimal places) |
25-O-methylkaravilagenin D |
-6.53 |
-7.30 |
-7.40 |
Kuguacin J |
-6.27 |
-6.73 |
-7.23 |
|
Kuguacin S |
-6.27 |
-7.33 |
-7.53 |
|
momordic acid |
-7.30 |
-7.70 |
-7.80 |
|
Momordicine I |
-5.93 |
-6.70 |
-7.17 |
References:
[1]Li C, Wen R, Liu D, Yan L, Gong Q, Yu H. Assessment of the Potential of Sarcandra glabra (Thunb.) Nakai. in Treating Ethanol-Induced Gastric Ulcer in Rats Based on Metabolomics and Network Analysis. Front Pharmacol. 2022 Jul 12;13:810344. doi: 10.3389/fphar.2022.810344. PMID: 35903344; PMCID: PMC9315220.
[2]Baikerikar S. Curcumin and Natural Derivatives Inhibit Ebola Viral Proteins: An In silico Approach. Pharmacognosy Res. 2017 Dec;9(Suppl 1):S15-S22. doi: 10.4103/pr.pr_30_17. PMID: 29333037; PMCID: PMC5757320.
Comments 10 : Figure 5 (Molecular Dynamics Simulation Results) presents RMSD and RMSF values, which are commonly used to assess stability. However, the discussion does not explain what an acceptable range for these values is or how these values compare to well-established ligand-protein interactions. Including a reference docking study for context would strengthen the findings.
Response 10 : We are very grateful for your valuable comments. We have reviewed a large number of research literatures and gained a deeper understanding of the standards related to the Root Mean Square Deviation (RMSD) and the Root Mean Square Fluctuation (RMSF). RMSD and RMSF are commonly used indicators for evaluating the stability of proteins [1]. RMSD is used to assess the stability of complex systems, and lower values indicate a more stable system [2]. Some studies have found that structures with a lower RMSD are more likely to become suitable candidate structures, and a model with an RMSD of 4.5 Å obviously corresponds to a good prediction [3, 4]. In our study, both the AKT1-momordic acid and IL6-momordic acid complex systems reached equilibrium after 10 ns and finally fluctuated around 3.2 Å and 2.5 Å. Therefore, the momordic acid small molecule exhibited high stability when binding to the AKT1 and IL6 target proteins.
RMSF can represent the flexibility of amino acid residues in proteins, which characterizes the fluctuation of each amino acid. This value indicates the stability of amino acids. The larger the RMSF value, the less stable the amino acid is, and conversely, the smaller the RMSF value, the more stable the amino acid is [5]. In our study, the RMSF values of the AKT1-momordic acid and IL6-momordic acid complex systems were relatively low (mostly below 4 Å), indicating low flexibility and high stability.
In addition, relevant studies have shown that the Radius of Gyration (Rg) characterizes the compactness of proteins. The lower this value is, the more compact the protein structure is, and the larger the value is, the looser the protein structure is [6]; the larger the Solvent Accessible Surface Area (SASA) value is, the larger the exposed area of the protein is and the fewer the buried hydrophobic surfaces are; the smaller the SASA value is, the more compact the protein structure is and the more buried hydrophobic surfaces there are [7]. In our experimental study, the Rg and SASA of the AKT1-momordic acid and IL6-momordic acid complex systems fluctuated relatively stably during the movement process, indicating that the AKT1-momordic acid and IL6-momordic acid complex systems remained stable and compact throughout the simulation process. The above-mentioned content has been supplemented at the corresponding positions in the manuscript. (lines 209-243, page 10)
References:
[1]Nettey-Oppong EE, Muhammad R, Ali A, Jeong HW, Seok YS, Kim SW, Choi SH. The Impact of Temperature and Pressure on the Structural Stability of Solvated Solid-State Conformations of Bombyx mori Silk Fibroins: Insights from Molecular Dynamics Simulations. Materials (Basel). 2024 Nov 21;17(23):5686. doi: 10.3390/ma17235686. PMID: 39685120; PMCID: PMC11642577.
[2]Feng MC, Luo F, Huang LJ, Li K, Chen ZM, Li H, Yao C, Qin BJ, Chen GZ. Rheum palmatum L. and Salvia miltiorrhiza Bge. Alleviates Acute Pancreatitis by Regulating Th17 Cell Differentiation: An Integrated Network Pharmacology Analysis, Molecular Dynamics Simulation and Experimental Validation. Chin J Integr Med. 2024 May;30(5):408-420. doi: 10.1007/s11655-023-3559-6. Epub 2023 Oct 20. PMID: 37861962.
[3]Ramezani M, Shamsara J. A Cross-Docking Study on Matrix Metalloproteinase Family. Antiinflamm Antiallergy Agents Med Chem. 2015;14(3):164-71. doi: 10.2174/1871523014666151020095718. PMID: 26872606.
[4]Hajdin CE, Ding F, Dokholyan NV, Weeks KM. On the significance of an RNA tertiary structure prediction. RNA. 2010 Jul;16(7):1340-9. doi: 10.1261/rna.1837410. Epub 2010 May 24. PMID: 20498460; PMCID: PMC2885683.
[5]Habibian-Dehkordi S, Farhadian S, Ghasemi M, et al. Insight into the binding behavior, structure, and thermal stability properties of β-lactoglobulin/Amoxicillin complex in a neutral environment[J]. Food Hydrocolloids, 2022, 133: 107830.
[6]Wang B L, Kou S B, Lin Z Y, et al. Investigation on the binding behavior between BSA and lenvatinib with the help of various spectroscopic and in silico methods[J]. Journal of Molecular Structure, 2020, 1204: 127521.
[7]Zhang X, Lu Y, Zhao R, et al. Study on simultaneous binding of resveratrol and curcumin to β-lactoglobulin: Multi-spectroscopic, molecular docking and molecular dynamics simulation approaches[J]. Food Hydrocolloids, 2022, 124: 107331.
Comments 11 : The conclusion states that momordic acid has the strongest interaction with AKT1 and IL6, but there is no detailed structural explanation for why this compound outperforms the others. A brief discussion on structure-activity relationships (SAR) and the molecular features that enhance its binding efficiency would improve the mechanistic understanding.
Response 11: In response to this question, we have added a relevant discussion to the discussion section:
"Notably, the superior binding affinity of momordic acid to AKT1 (-6.9 kcal/mol) and IL6 (-7.5 kcal/mol) aligns with structural insights from SAR studies on cucurbitane derivatives [1], where hydroxyl and carboxyl moieties enhance kinase interactions through hydrogen bonding with catalytic residues like ARG48 in AKT1 [2]. "
These structural elucidations align with recent advancements in multitarget drug design, where balanced polarity and conformational adaptability are recognized as critical for engaging cross-talk pathways like PI3K-AKT. We believe these additions significantly strengthen the mechanistic framework of the study. Thank you again for the constructive critique. (lines 258-266, page 11)
References:
[1]Cai, Xueyi, Ziqiang Lin, Qihong Zheng, Mengzhen Liao, Hongfu Li, Hanxiao Feng, Huan Chen, Yang Zhang, Xiao Chen, and Dong Liang. "Major Bitter-Tasting Compounds from the Dichloromethane Fraction of Bitter Gourd (Fruit of Momordica Charantia L.) Extract and Their Precursors." Journal of Agricultural and Food Chemistry 72, no. 40 (2024): 22237-49.
[2]Lee, Y. H., S. Y. Yoon, J. Baek, S. J. Kim, J. S. Yu, H. Kang, K. S. Kang, S. J. Chung, and K. H. Kim. "Metabolite Profile of Cucurbitane-Type Triterpenoids of Bitter Melon (Fruit of Momordica Charantia) and Their Inhibitory Activity against Protein Tyrosine Phosphatases Relevant to Insulin Resistance." J Agric Food Chem 69, no. 6 (2021): 1816-30.
Comments 12 : The Gene Ontology (GO) and KEGG enrichment analysis identifies multiple pathways related to T2DM, but it is unclear whether a false discovery rate (FDR) correction was applied. Since pathway analysis can generate many false positives, were any statistical thresholds (e.g., p-value cutoffs, q-values) used to determine significance? This clarification would help assess the robustness of the pathway predictions.
Response 12 : Thank you for your pertinent suggestions. During the GO and KEGG enrichment analyses, we used P-Value < 0.05 and corrected it with FDR. Then we screened and obtained the top 10 pathway information of BP (Biological Process), CC (Cellular Component), MF (Molecular Function) and the top 30 pathway information of KEGG. For specific content, please refer to Supplementary Table 1 - 4.
Comments 13 : The discussion should expand on the potential clinical applications of these findings. If cucurbitane-type triterpenoids are promising therapeutic candidates, how could they be formulated into treatments? Would they be best suited as dietary supplements, pharmaceutical drugs, or adjunct therapies? Providing a translational perspective will increase the study’s real-world relevance.
Response 13 : We sincerely appreciate the reviewer’s insightful suggestion,and we have had some discussion about the potential applications of cucurbitane-type triterpenoids. (lines 277-293, page 11)
"From a translational standpoint, cucurbitane-type triterpenoids physicochemical properties and bioavailability will dictate their therapeutic formulation. Given their natural origin and historical use in traditional diets, these compounds could be optimized as standardized dietary supplements to enhance glycemic control in prediabetic or early-stage T2DM populations. For instance, bioactive fractions of M. charantia have already been commercialized as nutraceuticals in Asian markets [1,2], though efficacy standardization remains a challenge. However, bioavailability challenges inherent to triterpenoids—as seen with berberine and curcumin [3]—necessitate formulation innovations. Structural optimization through glycosylation (to enhance solubility) or nanoparticle encapsulation (to improve intestinal absorption) could transform these compounds into pharmaceutical-grade agents, though rigorous safety assessments would be essential."
References:
[1]Wang, S., Z. Li, G. Yang, C. T. Ho, and S. Li. "Momordica Charantia: A Popular Health-Promoting Vegetable with Multifunctionality." Food Funct 8, no. 5 (2017): 1749-62.
[2]Jia, Shuo, Mingyue Shen, Fan Zhang, and Jianhua Xie. "Recent Advances in Momordica Charantia: Functional Components and Biological Activities." International Journal of Molecular Sciences 18, no. 12 (2017).
[3]Cicero, Arrigo F. G., and Alessandra Baggioni. "Berberine and Its Role in Chronic Disease." In Anti-Inflammatory Nutraceuticals and Chronic Diseases, 27-45, 2016.
Comments 14 : Some references are outdated (e.g., Sone et al., 2000). Since network pharmacology and molecular docking techniques have advanced significantly in recent years, citing more recent studies (2022-2024) on similar methodologies would enhance the study's credibility and relevance.
Response 14 : We sincerely thank the reviewers for their constructive feedback on updating the references related to network pharmacology and molecular docking methods. We have conducted a comprehensive inspection and revision of the use of references to enhance the credibility and relevance of this study.
Comments 15 : The manuscript states that Momordica charantia extracts have been shown to reduce blood glucose levels in patients, but no specific clinical trials are cited. If such evidence exists, a direct citation of human clinical trial data would strengthen the claim.
Response 15 : The hypoglycemic potential of M. charantia has been substantiated by clinical trials involving human participants, reinforcing its traditional use in diabetes management. A systematic review and meta-analysis of ten randomized controlled trials (n=1,045) demonstrated that M. charantia monotherapy significantly reduced fasting plasma glucose (−0.72 mmol/L), postprandial glucose (−1.43 mmol/L), and HbA1c (−0.26%) compared to placebo, with sustained effects over 4–16 weeks [1]. Further supporting this, a pilot clinical study reported that adjunctive use of 200 mL/day fresh M. charantia juice alongside conventional antidiabetic drugs reduced fasting and postprandial blood glucose by 30% and 32%, respectively, in type 2 diabetic patients over 90 days, outperforming pharmacotherapy alone [2]. These findings align with mechanistic studies suggesting insulinomimetic activity through enhanced β-cell regeneration and glucose uptake modulation. While variations in preparation methods (e.g., fresh vs. dried extracts) may influence efficacy, the cumulative clinical evidence positions M. charantia as a viable adjunct for glycemic control. (lines 52-67, page 2)
References:
[1]Peter, Emanuel L., Félicien Mushagalusa Kasali, Serawit Deyno, Andrew Mtewa, Prakash B. Nagendrappa, Casim Umba Tolo, Patrick Engeu Ogwang, and Duncan Sesaazi. "Momordica Charantia L. Lowers Elevated Glycaemia in Type 2 Diabetes Mellitus Patients: Systematic Review and Meta-Analysis." Journal of Ethnopharmacology 231 (2019): 311-24.
[2]Rauniyar, G. P., R. Sinha, K. Chapagain, R. Maskey, and D. R. Pandey. "Effects of Momordica Charantia (Karela/Bitterguord) in Type 2 Diabetic Patients Taking Allopathic Drugs: A Pilot Study." Kathmandu University Medical Journal 19, no. 2 (2021): 243-47.

Reviewer 2 Report
Comments and Suggestions for Authors
Title: Comprehensive Studies on the Regulation of Type 2 Diabetes by Cucurbitane-Type Triterpenoids in Momordica charantia L.: Insights from Network Pharmacology, Molecular Docking and Dynamics
This manuscript is well-written by the authors. I do believe that if they can improve the manuscripts following all comments. It might have a chance to publish in the journal.
Comments
- Line 6: Please add a comma after the name “Peihang Li”.
- Line 7: Abstract: The abstract form should be followed a format of the Journal. Please delete (1) Background:, (2) Methods:, (3) Results:, (4) Conclusions:.
- Abstract: It would be better if the authors present some quantitative results in the abstract.
- Line 29: Please delete a symbol “;” after the word “molecular dynamics simulation;”.
- Line 32: Please use the word “Momordica charantia L. (KG)” instead of KG. By the way, Momordica charantia L. (M. charantia) is better. I suggest to use the word “M. charantia” instead of KG in the whole manuscript.
- I suggest the authors to modify or re-write the introduction. What distinguishes your research between other? please give state of the art of your research
-The first paragraph: describe the importance or situation of Type 2 Diabetes in human.
-The second paragraph: describe Regulation of Type 2 Diabetes as well as the problem drugs/treatment for the regulation of Type 2 diabetes.
-The third paragraph: describe the information Momordica charantia. The authors should describe the traditional use, biological activities, and a network Pharmacology. The authors should describe molecular docking and dynamics simulation as tools to study insight into the mechanisms of compounds.
-The fourth paragraph: describe the objective of this study. Why the authors are interested in this study?
- Do the authors perform a statistical analysis? Please add the information.
- Figure 1: Please write the names of each compound under each Figure. If the Figure 1 is lactated to Table 1, the authors should use the same code. It will be easy to understand by readers.
- Please increase a magnification of all Figures.
- Discussion: Try to compare the results (the author’s hypothesis) with other finding by other researchers.
- Please delete some introduction and result sentences in the discussion part.
- Please check a journal format of references. Actually, the format is “Type 2 diabetes [1]” but the authors use “Type 2 diabetes[1]”. Hence, please correct and edit.
Author Response
Comments 1 : Line 6: Please add a comma after the name “Peihang Li”.
Response 1 : Thank you for your meticulous reminder. We have already made the revisions in the manuscript.
Comments 2 : Line 7: Abstract: The abstract form should be followed a format of the Journal. Please delete (1) Background:, (2) Methods:, (3) Results:, (4) Conclusions:.
Response 2 : Thank you for your meticulous reminder. We have strictly adhered to the journal's formatting guidelines and deleted the contents of the sections (1) Background:, (2) Methods:, (3) Results:, (4) Conclusions: from the manuscript. (lines 11-28, page 1)
Comments 3 : Abstract: It would be better if the authors present some quantitative results in the abstract.
Response 3 : Thank you very much, we have added some quantitative results in the abstract section as you suggested so that our readers can be clear about the findings of this thesis at first sight.(lines 11-28, page 1)
Comments 4 : Line 29: Please delete a symbol “;” after the word “molecular dynamics simulation;”.
Response 4 : Thank you for your meticulous reminder. We have already deleted the semicolon “;” after “molecular dynamics simulation;” in the manuscript. We are truly sorry for our carelessness in not noticing this detail.
Comments 5 : Line 32: Please use the word “Momordica charantia L. (KG)” instead of KG. By the way, Momordica charantia L. (M. charantia) is better. I suggest to use the word “M. charantia” instead of KG in the whole manuscript.
Response 5 : Thank you for your valuable suggestions. We acknowledge that using Momordica charantia L. (M. charantia) is an excellent recommendation, and we have already revised all the above-mentioned content in the manuscript.
Comments 6 : I suggest the authors to modify or re-write the introduction. What distinguishes your research between other? please give state of the art of your research
-The first paragraph: describe the importance or situation of Type 2 Diabetes in human.
-The second paragraph: describe Regulation of Type 2 Diabetes as well as the problem drugs/treatment for the regulation of Type 2 diabetes.
-The third paragraph: describe the information Momordica charantia. The authors should describe the traditional use, biological activities, and a network Pharmacology. The authors should describe molecular docking and dynamics simulation as tools to study insight into the mechanisms of compounds.
-The fourth paragraph: describe the objective of this study. Why the authors are interested in this study?
Response 6 : Thank you for providing us with such valuable suggestions and meticulous guidance. We have revised the introduction section, and the revised content is as follows. We hope to receive your positive feedback and that of the readers.
Type 2 Diabetes Mellitus (T2DM) has become a global health epidemic, affecting hundreds of millions of people worldwide and posing a significant challenge to public health systems[1]. The rising incidence of T2DM is largely attributed to modern lifestyles, including sedentary behaviors and unhealthy dietary habits, which have led to an increase in obesity and insulin resistance[2]. This condition not only impacts the quality of life of affected individuals but also places a substantial economic burden on healthcare systems due to the costs associated with long-term management and complications.
Momordica charantia L. (M. charantia), a cucurbitaceous plant widely consumed as both food and medicine, has garnered extensive scientific attention for its antidiabetic properties. Phytochemical analyses reveal its rich composition of cucurbitane-type triterpenoids (e.g., momordicine IV and charantoside B) [3], saponins (charantin) [4], peptides [5], and polysaccharides [6], which collectively contribute to its hypoglycemic effects. In vitro studies demonstrate that these compounds enhance glucose uptake in skeletal muscle cells via AMP-activated protein kinase (AMPK) activatio [7], inhibit α-glucosidase activity [8], and protect pancreatic β-cells from oxidative stress [9]. Animal models of streptozotocin-induced diabetes further validate these mechanisms, showing significant reductions in fasting blood glucose (FBG) and improved insulin sensitivity upon M. charantia extract administration [10, 11]. The hypoglycemic potential of M. charantia has been substantiated by clinical trials involving human participants, reinforcing its traditional use in diabetes management. A systematic review and meta-analysis of ten randomized controlled trials (n=1,045) demonstrated that M. charantia monotherapy significantly reduced fasting plasma glucose (−0.72 mmol/L), postprandial glucose (−1.43 mmol/L), and HbA1c (−0.26%) compared to placebo, with sustained effects over 4–16 weeks [12]. Further supporting this, a pilot clinical study reported that adjunctive use of 200 mL/day fresh M. charantia juice alongside conventional antidiabetic drugs reduced fasting and postprandial blood glucose by 30% and 32%, respectively, in type 2 diabetic patients over 90 days, outperforming pharmacotherapy alone[13]. These findings align with mechanistic studies suggesting insulinomimetic activity through enhanced β-cell regeneration and glucose uptake modulation. While variations in preparation methods (e.g., fresh vs. dried extracts) may influence efficacy, the cumulative clinical evidence positions M. charantia as a viable adjunct for glycemic control.
The regulation of T2DM involves a complex interplay of insulin signaling, inflammatory pathways (e.g., NF-κB), and metabolic regulators like PPARγ and PTP1B. While existing studies emphasize M. charantia's insulin secretagogue effects [14], emerging evidence suggests broader modulatory roles in lipid metabolism and adipokine regulation [15]. Recent proteomic analyses identify novel targets, including adiponectin upregulation and TNF-α suppression [16], positioning M. charantia as a multitarget therapeutic agent. However, the structural basis of its bioactive components interacting with key targets remains underexplored, particularly regarding the dynamic stability of ligand-receptor complexes.
This study aims to elucidate the multitarget effects of cucurbitane-type triterpenoids through an integrative approach combining network pharmacology, molecular docking, and molecular dynamics simulations.By mapping the interactions between M. charantia's active constituents (e.g., momordicoside L and charantin) and critical T2DM targets (PPARγ, PTP1B, and AKT1), we provide mechanistic insights into their synergistic hypoglycemic actions. Our novel computational framework not only validates hypothesized mechanisms but also reveals previously uncharacterized crosstalk between triterpenoid-induced AMPK activation and inflammatory pathway modulation. These findings advance the rational development of M. charantia-based nutraceuticals for precision diabetes management. (lines 33-84, page1~2)
References:
- Zimmet PZ. Diabetes and its drivers: the largest epidemic in human history? Clin Diabetes Endocrinol. 2017 Jan 18;3:1. doi: 10.1186/s40842-016-0039-3. PMID: 28702255; PMCID: PMC5471716.
- Sun H, Saeedi P, Karuranga S, Pinkepank M, Ogurtsova K, Duncan BB, Stein C, Basit A, Chan JCN, Mbanya JC, Pavkov ME, Ramachandaran A, Wild SH, James S, Herman WH, Zhang P, Bommer C, Kuo S, Boyko EJ, Magliano DJ. IDF Diabetes Atlas: Global, regional and country-level diabetes prevalence estimates for 2021 and projections for 2045. Diabetes Res Clin Pract. 2022 Jan;183:109119. doi: 10.1016/j.diabres.2021.109119. Epub 2021 Dec 6. Erratum in: Diabetes Res Clin Pract. 2023 Oct;204:110945. doi: 10.1016/j.diabres.2023.110945. PMID: 34879977; PMCID: PMC11057359.
- Cai X, Lin Z, Zheng Q, Liao M, Li H, Feng H, Chen H, Zhang Y, Chen X, Liang D. Major Bitter-Tasting Compounds from the Dichloromethane Fraction of Bitter Gourd (Fruit of Momordica charantia L.) Extract and Their Precursors. J Agric Food Chem. 2024 Oct 9;72(40):22237-22249. doi: 10.1021/acs.jafc.4c06506. Epub 2024 Sep 26. PMID: 39327224.
- Nuchtavorn N, Leanpolchareanchai J, Visansirikul S, Bunsupa S. Optimization of Magnetic and Paper-Based Molecularly Imprinted Polymers for Selective Extraction of Charantin in Momordica charantia. Int J Mol Sci. 2023 Apr 26;24(9):7870. doi: 10.3390/ijms24097870. PMID: 37175576; PMCID: PMC10178129.
- Yang B, Li X, Zhang C, Yan S, Wei W, Wang X, Deng X, Qian H, Lin H, Huang W. Design, synthesis and biological evaluation of novel peptide MC2 analogues from Momordica charantia as potential anti-diabetic agents. Org Biomol Chem. 2015 Apr 21;13(15):4551-61. doi: 10.1039/c5ob00333d. PMID: 25778708.
- Zhan, Kai, Xiaolong Ji, and Lei Luo. "Recent Progress in Research on Momordica Charantia Polysaccharides: Extraction, Purification, Structural Characteristics and Bioactivities." Chemical and Biological Technologies in Agriculture 10, no. 1 (2023).
- Kao PF, Cheng CH, Cheng TH, Liu JC, Sung LC. Therapeutic Potential of Momordicine I from Momordica charantia: Cardiovascular Benefits and Mechanisms. Int J Mol Sci. 2024 Sep 29;25(19):10518. doi: 10.3390/ijms251910518. PMID: 39408847; PMCID: PMC11477196.
- Poovitha S, Parani M. In vitro and in vivo α-amylase and α-glucosidase inhibiting activities of the protein extracts from two varieties of bitter gourd (Momordica charantia L.). BMC Complement Altern Med. 2016 Jul 18;16 Suppl 1(Suppl 1):185. doi: 10.1186/s12906-016-1085-1. PMID: 27454418; PMCID: PMC4959359.
- Singh J, Cumming E, Manoharan G, Kalasz H, Adeghate E. Medicinal chemistry of the anti-diabetic effects of momordica charantia: active constituents and modes of actions. Open Med Chem J. 2011;5(Suppl 2):70-7. doi: 10.2174/1874104501105010070. Epub 2011 Sep 9. PMID: 21966327; PMCID: PMC3174519.
- Wang Q, Wu X, Shi F, Liu Y. Comparison of antidiabetic effects of saponins and polysaccharides from Momordica charantia L. in STZ-induced type 2 diabetic mice. Biomed Pharmacother. 2019 Jan;109:744-750. doi: 10.1016/j.biopha.2018.09.098. Epub 2018 Nov 5. PMID: 30551527.
- Mahmoud MF, El Ashry FE, El Maraghy NN, Fahmy A. Studies on the antidiabetic activities of Momordica charantia fruit juice in streptozotocin-induced diabetic rats. Pharm Biol. 2017 Dec;55(1):758-765. doi: 10.1080/13880209.2016.1275026. PMID: 28064559; PMCID: PMC6130663.
- Peter EL, Kasali FM, Deyno S, Mtewa A, Nagendrappa PB, Tolo CU, Ogwang PE, Sesaazi D. Momordica charantia L. lowers elevated glycaemia in type 2 diabetes mellitus patients: Systematic review and meta-analysis. J Ethnopharmacol. 2019 Mar 1;231:311-324. doi: 10.1016/j.jep.2018.10.033. Epub 2018 Oct 30. PMID: 30385422.
- Rauniyar GP , Sinha R , Chapagain K , Maskey R , Pandey DR . Effects of Momordica Charantia (Karela/bitterguord) in Type 2 Diabetic Patients Taking Allopathic Drugs: A pilot study. Kathmandu Univ Med J (KUMJ). 2021 Apr-Jun;19(74):143-147. PMID: 34819444.
- Keller AC, Ma J, Kavalier A, He K, Brillantes AM, Kennelly EJ. Saponins from the traditional medicinal plant Momordica charantia stimulate insulin secretion in vitro. Phytomedicine. 2011 Dec 15;19(1):32-7. doi: 10.1016/j.phymed.2011.06.019. Epub 2011 Nov 30. PMID: 22133295; PMCID: PMC3389550.
- Zhang F, Zhang X, Yu J, Tan Y, Guo P, Wu C. The gut microbiota confers the lipid-lowering effect of bitter melon (Momordica charantia L.) In high-fat diet (HFD)-Induced hyperlipidemic mice. Biomed Pharmacother. 2020 Nov;131:110667. doi: 10.1016/j.biopha.2020.110667. Epub 2020 Aug 28. PMID: 32861068.
Comments 7 : Do the authors perform a statistical analysis? Please add the information.
Response 7 : Thank you for your meticulous reminder. In our research, relevant screening criteria were indeed involved during the process of data collection and processing. The data and criteria involved have been described in the revised manuscript and supplementary materials.
Comments 8 : Figure 1: Please write the names of each compound under each Figure. If the Figure 1 is lactated to Table 1, the authors should use the same code. It will be easy to understand by readers.
Response 8 : Thank you for your meticulous reminder. Just as you said, Figure 1 indeed presents the results of the compounds listed in Table 1. We consider your suggestion extremely valuable and have already made the corresponding adjustments to the above-mentioned content in the manuscript. (lines 106, page 3)
Comments 9 : Please increase a magnification of all Figures.
Response 9 : Thank you for your reminder. We have adjusted the figures format and font size in accordance with the requirements of the journal and your suggestions. We hope to provide you and the readers with a better reading experience.
The revised figures:
Comments 10 : Discussion: Try to compare the results (the author’s hypothesis) with other finding by other researchers.
Response 10 : Thank you for the precious suggestions you've put forward. We've made modifications and improvements to the discussion section as per your advice and that of the other reviewers. The revised content is as follows. (lines 251-327, page 11~12)
Our investigation elucidates the multitarget therapeutic potential of Momordica charantia-derived cucurbitane-type triterpenoids in T2DM intervention, advancing previous research by integrating network pharmacology with dynamic conformational analyses. While earlier studies primarily focused on single-target interactions or phenomenological observations of M. charantia extracts [1, 2],our systems-level approach reveals coordinated modulation of PI3K-AKT and AGE-RAGE signalling—a finding corroborated by recent work demonstrating pathway crosstalk in insulin sensitization [3]. Notably, the superior binding affinity of momordic acid to AKT1 (-7.3 kcal/mol) and IL6 (-7.7 kcal/mol) aligns with structural insights from SAR studies on cucurbitane derivatives[4],where hydroxyl and carboxyl moieties enhance kinase interactions through hydrogen bonding with catalytic residues like ARG48 in AKT1 [5]. This mechanistic clarity addresses a critical gap in prior computational models limited to static docking , as our 50-ns molecular dynamics simulations confirm stable ligand-receptor complexes through both polar interactions and hydrophobic stabilization of sterically bulky regions—a dual binding mode absent in less potent analogs like kuguacin J.
The identification of IL6 as a key anti-inflammatory target introduces a paradigm shift from conventional views of its pro-diabetic role. While chronic IL6 elevation exacerbates insulin resistance [6], our data suggest context-dependent modulation where momordic acid's stabilization of IL6-AKT1 interactions may reconcile contradictory reports on cytokine function in metabolic regulation. This aligns with emerging evidence that low-grade inflammation modulation—rather than complete suppression—optimizes glycemic control [7]. Such nuanced immunometabolic regulation positions cucurbitane triterpenoids as superior to single-pathway inhibitors like metformin[8],which primarily targets hepatic gluconeogenesis without addressing inflammatory components [9].
From a translational standpoint, cucurbitane-type triterpenoids' physicochemical properties and bioavailability will dictate their therapeutic formulation.Given their natural origin and historical use in traditional diets, these compounds could be optimized as standardized dietary supplements to enhance glycemic control in prediabetic or early-stage T2DM populations.For instance, bioactive fractions of M. charantia have already been commercialized as nutraceuticals in Asian markets [10, 11], though efficacy standardization remains a challenge. However, bioavailability challenges inherent to triterpenoids—as seen with berberine and curcumin —necessitate formulation innovations[12]. Structural optimization through glycosylation or nanoparticle encapsulation (to improve intestinal absorption) could transform these compounds into pharmaceutical-grade agents, though rigorous safety assessments would be essential. Preclinical models demonstrate feasibility: saponin-rich M. charantia fractions reduced fasting glucose in diabetic mice via AMPK activation [13], while cucurbitane glycosides increased GLUT4 translocation in adipocytes [14]. These findings validate our computational predictions and suggest prioritized evaluation of momordic acid in rodent models using dose-escalation protocols mirroring successful anti-diabetic natural product development [15, 16].
Methodologically, our integration of KEGG pathway analysis with molecular dynamics represents a significant advancement over previous network pharmacology studies on M. charantia [17]. While Taheri et al. identified PI3K/AKT dysregulation in insulin resistance, our work uniquely maps ligand-induced conformational stabilization to pathway reactivation—a critical step for rational drug design.Notably, our findings complement recent discoveries of isomerization phenomena in cucurbitane triterpenoids during extraction,emphasizing the necessity of dynamic conformational analysis.Future studies should employ surface plasmon resonance to experimentally validate binding affinities, particularly for momordic acid's interaction with AKT1's PH domain, where its flexible side chains occupy a hydrophobic pocket adjacent to the catalytic cleft. Such validation would bridge the gap between computational prediction and therapeutic application, addressing a persistent limitation in phytochemical research [18].
In conclusion, this study establishes a structural and systems-level framework for repurposing cucurbitane triterpenoids as multitarget T2DM therapeutics. By elucidating both molecular interactions and pathway synergies, we provide a roadmap for developing standardized nutraceuticals or optimized pharmaceuticals. Clinical trials should explore adjunctive use with existing therapies (e.g., metformin or GLP-1 agonists), leveraging these compounds' unique capacity to concurrently target insulin signalling and inflammation—a combinatorial strategy poised to address diabetes-associated comorbidities more effectively than current monotherapies.
Despite these advances, our study has limitations. While computational models predict therapeutic potential, in vivo validation is imperative to confirm bioavailability, tissue-specific effects, and long-term safety.Particular attention should be paid to the dose-dependent effects observed in STZ-induced diabetic mice models, as well as potential microbiota-mediated mechanisms reported in HFD-fed mice. Future work should employ diabetic rodent models to assess dose-dependent glucose-lowering effects and compare cucurbitane-type triterpenoids with existing antidiabetic agents. Additionally, clinical trials could evaluate synergistic effects when these compounds are co-administered with metformin or GLP-1 agonists, leveraging their multitarget profiles to enhance therapeutic outcomes. These interactions can also be confirmed experimentally using techniques such as surface plasmon resonance (SPR) or enzyme inhibition analysis. These methods can provide direct measurements of combined affinity and kinetics, which will further validate computational predictions and strengthen the credibility of the findings.
References:
[1]Jiang, B., M. Ji, W. Liu, L. Chen, Z. Cai, Y. Zhao, and X. Bi. "Antidiabetic Activities of a Cucurbitane‑Type Triterpenoid Compound from Momordica Charantia in Alloxan‑Induced Diabetic Mice." Mol Med Rep 14, no. 5 (2016): 4865-72.
[2]Bortolotti, Massimo, Daniele Mercatelli, and Letizia Polito. "Momordica Charantia, a Nutraceutical Approach for Inflammatory Related Diseases." Frontiers in Pharmacology 10 (2019).
[3]Taheri, R., Y. Mokhtari, A. M. Yousefi, and D. Bashash. "The Pi3k/Akt Signaling Axis and Type 2 Diabetes Mellitus (T2dm): From Mechanistic Insights into Possible Therapeutic Targets." Cell Biol Int 48, no. 8 (2024): 1049-68.
[4]Cai, Xueyi, Ziqiang Lin, Qihong Zheng, Mengzhen Liao, Hongfu Li, Hanxiao Feng, Huan Chen, Yang Zhang, Xiao Chen, and Dong Liang. "Major Bitter-Tasting Compounds from the Dichloromethane Fraction of Bitter Gourd (Fruit of Momordica Charantia L.) Extract and Their Precursors." Journal of Agricultural and Food Chemistry 72, no. 40 (2024): 22237-49.
[5]Lee, Y. H., S. Y. Yoon, J. Baek, S. J. Kim, J. S. Yu, H. Kang, K. S. Kang, S. J. Chung, and K. H. Kim. "Metabolite Profile of Cucurbitane-Type Triterpenoids of Bitter Melon (Fruit of Momordica Charantia) and Their Inhibitory Activity against Protein Tyrosine Phosphatases Relevant to Insulin Resistance." J Agric Food Chem 69, no. 6 (2021): 1816-30.
[6]Htwe, Thae Nu. "Metabolic Risk Markers in Insulin Resistance and Non-Insulin Resistance Type 2 Diabetes Mellitus." SOJ Diabetes and Endocrinology Care 1, no. 2 (2021).
[7]Geyer, Natalie. Targeting the Hedgehog and Pi3k/Akt/Mtor Signaling Pathways in Rhabdomyosarcoma. Dissertation, Göttingen, Georg-August Universität, 2018, 2018.
[8]Tong, Xiaolin, Jia Xu, Fengmei Lian, Xiaotong Yu, Yufeng Zhao, Lipeng Xu, Menghui Zhang, Xiyan Zhao, Jian Shen, Shengping Wu, Xiaoyan Pang, Jiaxing Tian, Chenhong Zhang, Qiang Zhou, Linhua Wang, Bing Pang, Feng Chen, Zhiping Peng, Jing Wang, Zhong Zhen, Chao Fang, Min Li, Limei Chen, Liping Zhao, Elisabeth M. Bik, Martin J. Blaser, Jun Wang, and Xinhua Xiao. "Structural Alteration of Gut Microbiota During the Amelioration of Human Type 2 Diabetes with Hyperlipidemia by Metformin and a Traditional Chinese Herbal Formula: A Multicenter, Randomized, Open Label Clinical Trial." mBio 9, no. 3 (2018).
[9]Madiraju, Anila K., Derek M. Erion, Yasmeen Rahimi, Xian-Man Zhang, Demetrios T. Braddock, Ronald A. Albright, Brett J. Prigaro, John L. Wood, Sanjay Bhanot, Michael J. MacDonald, Michael J. Jurczak, Joao-Paulo Camporez, Hui-Young Lee, Gary W. Cline, Varman T. Samuel, Richard G. Kibbey, and Gerald I. Shulman. "Metformin Suppresses Gluconeogenesis by Inhibiting Mitochondrial Glycerophosphate Dehydrogenase." Nature 510, no. 7506 (2014): 542-46.
[10]Wang, S., Z. Li, G. Yang, C. T. Ho, and S. Li. "Momordica Charantia: A Popular Health-Promoting Vegetable with Multifunctionality." Food Funct 8, no. 5 (2017): 1749-62.
[11]Jia, Shuo, Mingyue Shen, Fan Zhang, and Jianhua Xie. "Recent Advances in Momordica Charantia: Functional Components and Biological Activities." International Journal of Molecular Sciences 18, no. 12 (2017).
[12]Cicero, Arrigo F. G., and Alessandra Baggioni. "Berberine and Its Role in Chronic Disease." In Anti-Inflammatory Nutraceuticals and Chronic Diseases, 27-45, 2016.
[13]Shih, Chun‐Ching, Min‐Tzong Shlau, Cheng‐Hsiu Lin, and Jin‐Bin Wu. "Momordica Charantia Ameliorates Insulin Resistance and Dyslipidemia with Altered Hepatic Glucose Production and Fatty Acid Synthesis and Ampk Phosphorylation in High‐Fat‐Fed Mice." Phytotherapy Research 28, no. 3 (2013): 363-71.
[14]Han, Joo‐Hui, Nguyen Quoc Tuan, Min‐Ho Park, Khong Trong Quan, Joonseok Oh, Kyung‐Sun Heo, MinKyun Na, and Chang‐Seon Myung. "Cucurbitane Triterpenoids from the Fruits of Momordica Charantia Improve Insulin Sensitivity and Glucose Homeostasis in Streptozotocin‐Induced Diabetic Mice." Molecular Nutrition & Food Research 62, no. 7 (2018).
[15]Jiang, Shu-Jun. "Berberine Inhibits Hepatic Gluconeogenesisviathe Lkb1-Ampk-Torc2 Signaling Pathway in Streptozotocin-Induced Diabetic Rats." World Journal of Gastroenterology 21, no. 25 (2015).
[16]Liu, Xiaojia, Mingxiao Yin, Jingwen Dong, Genxiang Mao, Wenjian Min, Zean Kuang, Peng Yang, Lu Liu, Na Zhang, and Hongbin Deng. "Tubeimoside-1 Induces Tfeb-Dependent Lysosomal Degradation of Pd-L1 and Promotes Antitumor Immunity by Targeting Mtor." Acta pharmaceutica sinica B 11, no. 10 (2021): 3134-49.
[17]Rivera, R. G., Jr., P. J. S. Regidor, E. C. Ruamero, C. D. R. Delos Santos, C. B. Gomez, E. J. V. Allanigue, and M. V. Salinas. "Applying Network Pharmacology and Molecular Docking in the Screening for Molecular Mechanisms of Ampalaya (Momordica Charantia L.) and Banaba (Lagerstroemia Speciosa L.) against Type 2 Diabetes Mellitus." Acta Med Philipp 58, no. 8 (2024): 108-24.
[18]Varadi, Mihaly, Maxim Tsenkov, and Sameer Velankar. "Challenges in Bridging the Gap between Protein Structure Prediction and Functional Interpretation." Proteins: Structure, Function, and Bioinformatics 93, no. 1 (2023): 400-10.
Comments 11 : Please delete some introduction and result sentences in the discussion part.
Response 11 : Thank you for your valuable suggestions. We have revised the discussion section as per your and other reviewers' recommendations. We hope the revised manuscript will meet with your approval and that of the readers.
Comments 12 : Please check a journal format of references. Actually, the format is “Type 2 diabetes [1]” but the authors use “Type 2 diabetes[1]”. Hence, please correct and edit.
Response 12 : Thank you for your meticulous reminders and valuable suggestions. We have made comprehensive revisions to the references in full compliance with the journal's formatting requirements.

Reviewer 3 Report
Comments and Suggestions for Authors
I have thoroughly reviewed the manuscript titled Comprehensive Studies on the Regulation of Type 2 Diabetes by Cucurbitane-type Triterpenoids in Momordica charantia L.: Insights from Network Pharmacology, Molecular Docking, and Dynamics. This manuscript has the potential to be published in Pharmaceuticals; however, the authors must first address the questions and respond to the recommendations provided.
1. In the Introduction section, the full scientific name of the key plant discussed in the article, Momordica charantia L., should be stated correctly when first mentioned in a paragraph or section. It is important to follow the proper conventions for writing scientific plant names.
2. Additionally, there are instances in the manuscript where the scientific name is not written correctly, such as in the table title of Table 1 and on lines 151–152. These should be revised to ensure consistency and adherence to scientific writing standards.
3. The Introduction section contains insufficient literature review on relevant studies, particularly regarding the investigation of plant extracts in in vitro, in vivo, and clinical studies. Additionally, there is a lack of information on the phytochemicals found in this plant. Including these details would enhance the significance of the study and provide readers with a better understanding of Momordica charantia L..
4. The in-text citations do not conform to the journal’s referencing guidelines. Please review and adjust the references to ensure they align with the journal’s formatting requirements.
5. Figures 2–5 are too small, making it difficult to see the details clearly. Please consider enlarging them to improve readability and ensure that all important elements are visible.
6. The Discussion section contains limited comparisons between the study’s findings and previous research, which reduces its impact. To enhance the discussion, it is recommended to include more critical analysis and comparisons with existing literature. Additionally, further discussion on the identification of candidate compounds affecting Type 2 Diabetes should be included, covering aspects such as plant parts used, extraction methods, and relevant experimental studies. This will strengthen the study’s significance and provide a more comprehensive interpretation of the results.
There are minor spelling and grammatical errors throughout the manuscript. The authors should carefully proofread the text to ensure accuracy and adherence to proper English grammar.
Author Response
Comments 1 : In the Introduction section, the full scientific name of the key plant discussed in the article, Momordica charantia L., should be stated correctly when first mentioned in a paragraph or section. It is important to follow the proper conventions for writing scientific plant names.
Response 1 : Thank you for your valuable suggestions. Throughout this article, our main research focuses on the plant Momordica charantia L. of the genus Momordica in the Cucurbitaceae family. We have already added the complete scientific name of the plant in a standardized way in the Introduction section. (lines 41-42, page 2)
Comments 2 : Additionally, there are instances in the manuscript where the scientific name is not written correctly, such as in the table title of Table 1 and on lines 151–152. These should be revised to ensure consistency and adherence to scientific writing standards.
Response 2 : Thank you for your valuable suggestions. We have conducted a comprehensive review of the manuscript content and made revisions in accordance with your recommendations and scientific writing standards. We hope the revised manuscript will meet with your approval and that of the readers.
Comments 3 : The Introduction section contains insufficient literature review on relevant studies, particularly regarding the investigation of plant extracts in in vitro, in vivo, and clinical studies. Additionally, there is a lack of information on the phytochemicals found in this plant. Including these details would enhance the significance of the study and provide readers with a better understanding of Momordica charantia L..
Response 3 : Thank you for your meticulous reminder. Firstly, we have read a large number of relevant research literatures and supplemented them in the Introduction section of the article. We have systematically explored the effects of Momordica charantia on type 2 diabetes mellitus (T2DM) in in vitro, in vivo, and clinical studies. Relevant studies have shown that Momordica charantia can lower blood sugar and improve T2DM by activating AMP-activated protein kinase (AMPK) [1], inhibiting the activity of α-glucosidase [2], and protecting pancreatic β-cells from oxidative stress [3], among other ways. In animal experimental studies, after administering the extract of Momordica charantia, the fasting blood glucose (FBG) significantly decreased, and insulin sensitivity was improved. In clinical studies, after T2DM patients took the extract of Momordica charantia, the blood glucose-related indicators could be improved, the fasting blood glucose and postprandial blood glucose were reduced, and the quality of life of patients was enhanced. Secondly, we have reviewed the research progress of the chemical components of Momordica charantia, and presented the information of the potential components of the cucurbitane-type triterpenoid components of Momordica charantia acting on T2DM in the manuscript through tables and figures. For the specific content, please refer to Table 1 and Figure 1. Momordica charantia is a common vegetable in many places and has the effect of lowering blood sugar. It is rich in cucurbitane-type triterpenoids (such as momordicine IV and charantoside B) [4], saponins (charantin) [5], peptides [6], and polysaccharides [7]. It is of great significance to conduct in-depth research on the blood-sugar-lowering chemical components of Momordica charantia. Therefore, we focused on the cucurbitane-type triterpenoid components in Momordica charantia and carried out the research of this article. All of the above content has been supplemented and improved at the corresponding positions in the article. (lines 41-67, page 2)
References:
[1]Kao PF, Cheng CH, Cheng TH, Liu JC, Sung LC. Therapeutic Potential of Momordicine I from Momordica charantia: Cardiovascular Benefits and Mechanisms. Int J Mol Sci. 2024 Sep 29;25(19):10518. doi: 10.3390/ijms251910518. PMID: 39408847; PMCID: PMC11477196.
[2]Poovitha S, Parani M. In vitro and in vivo α-amylase and α-glucosidase inhibiting activities of the protein extracts from two varieties of bitter gourd (Momordica charantia L.). BMC Complement Altern Med. 2016 Jul 18;16 Suppl 1(Suppl 1):185. doi: 10.1186/s12906-016-1085-1. PMID: 27454418; PMCID: PMC4959359.
[3]Singh J, Cumming E, Manoharan G, Kalasz H, Adeghate E. Medicinal chemistry of the anti-diabetic effects of momordica charantia: active constituents and modes of actions. Open Med Chem J. 2011;5(Suppl 2):70-7. doi: 10.2174/1874104501105010070. Epub 2011 Sep 9. PMID: 21966327; PMCID: PMC3174519.
[4]Cai X, Lin Z, Zheng Q, Liao M, Li H, Feng H, Chen H, Zhang Y, Chen X, Liang D. Major Bitter-Tasting Compounds from the Dichloromethane Fraction of Bitter Gourd (Fruit of Momordica charantia L.) Extract and Their Precursors. J Agric Food Chem. 2024 Oct 9;72(40):22237-22249. doi: 10.1021/acs.jafc.4c06506. Epub 2024 Sep 26. PMID: 39327224.
[5]Nuchtavorn N, Leanpolchareanchai J, Visansirikul S, Bunsupa S. Optimization of Magnetic and Paper-Based Molecularly Imprinted Polymers for Selective Extraction of Charantin in Momordica charantia. Int J Mol Sci. 2023 Apr 26;24(9):7870. doi: 10.3390/ijms24097870. PMID: 37175576; PMCID: PMC10178129.
[6]Yang B, Li X, Zhang C, Yan S, Wei W, Wang X, Deng X, Qian H, Lin H, Huang W. Design, synthesis and biological evaluation of novel peptide MC2 analogues from Momordica charantia as potential anti-diabetic agents. Org Biomol Chem. 2015 Apr 21;13(15):4551-61. doi: 10.1039/c5ob00333d. PMID: 25778708.
[7]Zhan, Kai, Xiaolong Ji, and Lei Luo. "Recent Progress in Research on Momordica Charantia Polysaccharides: Extraction, Purification, Structural Characteristics and Bioactivities." Chemical and Biological Technologies in Agriculture 10, no. 1 (2023).
Comments 4 : The in-text citations do not conform to the journal’s referencing guidelines. Please review and adjust the references to ensure they align with the journal’s formatting requirements.
Response 4 : Thank you for your meticulous reminders. We have reviewed and revised all relevant content in the manuscript in full compliance with the journal's reference citation requirements.
Comments 5 : Figures 2–5 are too small, making it difficult to see the details clearly. Please consider enlarging them to improve readability and ensure that all important elements are visible.
Response 5 : Thank you for your valuable suggestions. We have adjusted the figures and fonts in the manuscript in accordance with the requirements of the journal and your suggestions. We hope to provide you and the readers with a better reading experience. At the same time, we apologize for the oversight on our part that resulted in the pictures being too small.
The revised figures:
Comments 6 : The Discussion section contains limited comparisons between the study’s findings and previous research, which reduces its impact. To enhance the discussion, it is recommended to include more critical analysis and comparisons with existing literature. Additionally, further discussion on the identification of candidate compounds affecting Type 2 Diabetes should be included, covering aspects such as plant parts used, extraction methods, and relevant experimental studies. This will strengthen the study’s significance and provide a more comprehensive interpretation of the results.
Response 6 : Thank you for your invaluable suggestions. We have revised and expanded the Discussion section as per your recommendations and those of other reviewers. In accordance with your suggestions and those of other reviewers, we have revised and expanded the Discussion section. Through literature review and network pharmacology approaches, we systematically screened 22 cucurbitane-type triterpenoid compounds from Momordica charantia and conducted an in-depth exploration of their mechanisms of action in intervening with Type 2 Diabetes Mellitus (T2DM). Among them, Kuguacin J, 25-O-methylkaravilagenin D, Momordicine I, momordic acid, and Kuguacin S became the focal points of our analysis. Previous studies have confirmed the effectiveness of the compound Momordicine I in improving T2DM [1-3]. In contrast, there is relatively less research on the other compounds. Additionally, AKT1 has been widely reported as a key regulatory factor in the pathogenesis of T2DM. These findings validate the reliability of the research framework of this study. Regarding the candidate compounds that affect T2DM, we have systematically organized the relevant data in Table 1 and Figure 1. It is worth noting that momordic acid demonstrated a better regulatory effect on T2DM in our study. However, there is still a scarcity of research on its sources from plant parts, extraction methods, and clinical applications. We believe that this will also be one of the key focuses of future research on the improvement of T2DM by Momordica charantia. These updates have been incorporated into the corresponding sections of the manuscript.
References:
[1]Keller AC, Ma J, Kavalier A, He K, Brillantes AM, Kennelly EJ. Saponins from the traditional medicinal plant Momordica charantia stimulate insulin secretion in vitro. Phytomedicine. 2011 Dec 15;19(1):32-7. doi: 10.1016/j.phymed.2011.06.019. Epub 2011 Nov 30. PMID: 22133295; PMCID: PMC3389550.
[2]Kashyap H, Gupta S, Bist R. Impact of Active Antihyperglycemic Components as Herbal Therapy for Preventive Health Care Management of Diabetes. Curr Mol Med. 2019;19(1):12-19. doi: 10.2174/1566524019666190219124301. PMID: 30806316.
[3]Wu SB, Yue GG, To MH, Keller AC, Lau CB, Kennelly EJ. Transport in Caco-2 cell monolayers of antidiabetic cucurbitane triterpenoids from Momordica charantia fruits. Planta Med. 2014 Jul;80(11):907-11. doi: 10.1055/s-0034-1382837. Epub 2014 Aug 12. PMID: 25116119.

Round 2
Reviewer 1 Report
Comments and Suggestions for Authors
Acceptable for publication.
Reviewer 3 Report
Comments and Suggestions for Authors
After reviewing the revised manuscript again, the author has responded to the suggestions and addressed the questions I raised very well. I believe this manuscript can be accepted for publication.